# Selenium-Fortified Kombucha–Pollen Beverage by In Situ Biosynthesized Selenium Nanoparticles with High Biocompatibility and Antioxidant Activity

**DOI:** 10.3390/antiox12091711

**Published:** 2023-09-02

**Authors:** Naomi Tritean, Ștefan-Ovidiu Dima, Bogdan Trică, Rusăndica Stoica, Marius Ghiurea, Ionuț Moraru, Anisoara Cimpean, Florin Oancea, Diana Constantinescu-Aruxandei

**Affiliations:** 1Bioresources, Polymers and Analysis Departments, National Institute for Research & Development in Chemistry and Petrochemistry—ICECHIM, Splaiul Independenței No. 202, Sector 6, 060021 Bucharest, Romania; naomi.tritean@icechim.ro (N.T.); ovidiu.dima@icechim.ro (Ș.-O.D.); bogdan.trica@icechim.ro (B.T.); rusandica.stoica@icechim.ro (R.S.); marius.ghiurea@icechim.ro (M.G.); 2Faculty of Biology, University of Bucharest, Splaiul Independentei No. 91-95, 050095 Bucharest, Romania; anisoara.cimpean@bio.unibuc.ro; 3Postdoctoral School, National University of Science and Technology Politehnica of Bucharest, Splaiul Independenței No. 313, 060042 Bucharest, Romania; 4Medica Laboratories, Str. Frasinului nr. 11, 075100 Otopeni, Romania; ionut.moraru@pro-natura.ro; 5Faculty of Biotechnologies, University of Agronomic Sciences and Veterinary Medicine of Bucharest, Mărăști Blv. No. 59, Sector 1, 011464 Bucharest, Romania

**Keywords:** biogenic nanoselenium, kombucha fermentation, selenium-nanoparticles-enriched bee bread, antioxidant, biocompatible, response surface methodology

## Abstract

Biogenic selenium nanoparticles (SeNPs) have been shown to exhibit increased bioavailability. Fermentation of pollen by a symbiotic culture of bacteria and yeasts (SCOBY/Kombucha) leads to the release of pollen content and enhances the prebiotic and probiotic effects of Kombucha. The aim of this study was to fortify Kombucha beverage with SeNPs formed in situ by Kombucha fermentation with pollen. Response Surface Methodology (RSM) was used to optimize the biosynthesis of SeNPs and the pollen-fermented Kombucha beverage. SeNPs were characterized by Transmission electron microscopy energy-dispersive X-ray spectroscopy (TEM-EDX), Fourier-transform infrared spectroscopy (FTIR), Dynamic light scattering (DLS), and Zeta potential. The pollen-fermented Kombucha beverage enriched with SeNPs was characterized by measuring the total phenolic content, antioxidant activity, soluble silicon, saccharides, lactic acid, and the total content of Se^0^. The polyphenols were identified by liquid chromatography–mass spectrometry (LC-MS). The pollen and the bacterial (nano)cellulose were characterized by scanning electron microscopy-energy dispersive X-ray spectroscopy (SEM-EDX), FTIR, and X-Ray diffraction (XRD). We also assessed the in vitro biocompatibility in terms of gingival fibroblast viability and proliferation, as well as the antioxidant activity of SeNPs and the pollen-fermented Kombucha beverage enriched with SeNPs. The results highlight their increased biological performance in this regard.

## 1. Introduction

The dietary supplements market is one of the most effervescent segments of the pharmaceutical market, with constant growth each year [1]. There is a growing public interest in natural plant and bee extracts due to their therapeutic and prophylactic potential based on their various biological activities [2,3,4,5,6]. 

Customers are more health-aware, which has led to a bloom in popularity of healthy beverages such as Kombucha—a tea-based fermented beverage produced by acetic acid bacteria (i.e., *Komagataeibacter* spp., *Gluconobacter* spp., and *Acetobacter* spp.), yeasts (i.e., *Zygosaccharomyces* spp. and *Brettanomyces* spp.), and lactobacilli (i.e., *Lactobacillus* spp., *Lactococcus* spp.) [7,8,9,10,11], which has the ability to promote symbiotic biofilm formation and to reduce inflammation and lipid peroxidation [12,13]. Together with the production of this functional beverage, on the surface of the culture media, the consortium forms a pellicle of bacterial cellulose (BC), which represents the common habitat of the microbial consortium that includes both anaerobically and aerobically growing populations [9,10,14]. Never-dried bacterial nanocellulose (NDBNC) has specific properties that result from its biosynthesis particularities. Today, a wide palette of BNC properties is known, like biocompatibility, high hydrophilicity, flexibility, transparency, high mechanical strength and chemical stability, high surface area, and rich surface chemistry [15,16]. Moreover, nanostructured bacterial nanocellulose membranes were demonstrated to be an effective carrier for anti-inflammatory and antioxidant bioactive ingredients [17]. 

The fermentation of pollen in the honeycombs and the resulting semisolid fermentation product—bee bread—has been known since ancient times. There are several reports about the increased bioavailability of bee bread in terms of improved bio-assimilation and, moreover, its rich nutrient composition, with a significant impact on human health [18,19,20,21]. Fermentation of pollen by a symbiotic culture of bacteria and yeasts (SCOBY/Kombucha) leads to the release of pollen content, including biosilica, and also enhances the prebiotic and probiotic effects of Kombucha [5].

Selenium nanoparticles (SeNPs), especially biogenic ones, have been shown to be active against dysbiotic biofilm and decrease inflammation, mainly due to their antioxidant activity [22,23,24,25,26,27,28,29,30,31,32,33]. Selenium (Se) has a very narrow physiological window, with the difference between the beneficial and toxic dose being less than an order of magnitude [34]. SeNPs have been shown to present lower toxicity than the inorganic Se salts such as sodium selenite and selenate in some cases, probably due to the slow release of soluble Se species [35,36,37]. 

In other cases, the effects were similar to sodium selenite, and it is possible that the effects depend on the synthesis route and properties of SeNPs [36,38]. Se exerts epigenetic effects due to activation of the enzyme betaine homocysteine methyltransferase (BMHT), a key enzyme in carbon metabolism and S-adenosyl methionine restoration [25,39]. Polyphenols, including those from tea (*Camellia sinensis*), synergize the biological activity of nanoselenium, prevent and disperse dysbiotic biofilms, and reduce inflammation [22,30].

SeNPs have not been produced so far with a Kombucha consortium, although both yeast and lactic acid bacteria, components of the SCOBY consortium, have been shown to produce biogenic SeNPs [40,41,42,43,44]. The aim of this study was to fortify and optimize Kombucha beverage with biocompatible and antioxidant SeNPs formed in situ by Kombucha fermentation with pollen.

## 2. Materials and Methods

### 2.1. Materials 

Fresh polyfloral pollen collected from Dâmbovița, Romania at the end of July was used in this work. Biocompatibility assays were performed on gingival fibroblasts (HGF-1, ATCC CRL-2014). The Symbiotic Culture of Bacteria and Yeast (SCOBY) was sourced from a Romanian culture [14]. The following materials were used: ascorbic acid, sodium selenite, 1,1,3,3-tetramethoxypropane 99%, 2-thiobarbituric acid ≥98%, gallic acid, quercetin dihydrate, neocuproine, 2,2-Diphenyl-1-picrylhydrazyl, Dulbecco’s Modified Eagle’s Medium-low glucose, D-(+)-Glucose, sodium bicarbonate, trypsin from porcine pancreas, dimethyl sulfoxide 99.5%, 4’,6-diamidino-2-phenyindole, dilactate sodium deoxycholate, antibiotic antimycotic solution 100× stabilized, catechin hydrate (Sigma Aldrich, St. Louis, MO, USA), Cell counting kit-8 (Bimake, Houston, TX, USA), Phalloidin-iFluor 488Reagent (Abcam, Cambridge, UK), absolute ethanol 99.5%, hydrochloric acid 37%, acetic acid (Chimopar Srl, Bucharest, Romania), Viability/Cytotoxicity Assay Kit (Biotium, Fremont, CA, USA), Trolox 97% (Acros Organics, Thermo Fisher Scientific, Pittsburgh, PA, USA), Folin–Ciocalteu’s phenol reagent, Iron chloride (III) (Merck, Darmstadt, Germany), disodium hydrogen phosphate dihydrate, sodium dihydrogen phosphate monohydrate, tris-(hydroxymethyl)-aminomethane, potassium iodide, sodium chloride, paraformaldehyde, Triton X-100, trichloroacetic acid for analysis, hydrogen peroxide 30%, hydrochloric acid 1N, sodium acetate, sodium hydroxide, sodium sulfide, sodium nitrite, sodium molybdate (Scharlau, Barcelona, Spain), 2,4,6-tri(2-pyridyl-1,3,5-triazine) 98%, sodium n-dodecyl sulfate 99%, aluminium chloride, hydroxylamine hydrochloride (Alfa Aesar, Haverhill, MA, USA), methanol (Honeywell, Wabash, IN, USA), albumin bovine fraction V, pH 7.0 (Janssen Chimica, Beerse, Belgium), FBS USDA APPD. ORIGIN (Thermo Fisher Scientific, Waltham, MA, USA), chlorogenic acid, 2,7-dichlorodihydrofluorescein (Cayman Chemicals, Ann Arbor, MI, USA), t-resveratrol (Dr. Ehrenstorfer GmbH, Germany of 99.8% purity), quercetin 3-rutinoside, trifluoroacetic acid (Sigma-Aldrich, St. Louis, MO, USA), for LC-MS and HPLC-DAD: acetic acid glacial, acetonitrile and methanol, HPLC Plus Gradient (CARLO ERBA Reagents S.A.S, Val de Reuil Cedex, France), and ethanol (Merck, Darmstadt, Germany).

### 2.2. Kombucha Beverage with SeNPs Formed In Situ by Kombucha Fermentation with Pollen 

#### 2.2.1. Response Surface Methodology (RSM)

Response Surface Methodology (RSM)–Central Composite Design (CCD) was used to optimize the biosynthesis of SeNPs by pollen fermentation with a Kombucha consortium, supplemented with selenium salt. The experimental design, randomization, and data analysis were performed using Design-Expert software version 11.0.5.0. After establishing the factors/parameters of the experimental design, i.e., SCOBY (mL), pollen (g), and sodium selenite (mg), an experimental domain was defined by selecting a range for each factor (Table 1). 

The program generated a total of 25 experiments, and each combination of factors was performed in duplicate. For the estimation of the standard deviation, 5 experiments were performed in the center of the model (Appendix A).

After adding 10 g/L black tea to previously sterilized double-distilled water heated to 95 °C, the tea was left to infuse for 5 min. Afterwards, 24 g of sugar was added to 720 mL autoclaved jars, over which 300 mL from the previously prepared black tea was poured and left to cool at room temperature. Subsequently, 50 g/L bacterial cellulose membrane was added in each jar [5]. The final step was the addition of SCOBY, pollen, and sodium selenite according to the experimental plan. The jars were covered with sterile gauze and kept at 30 °C for 14 days. The experimental variants were followed by a control, (K) in which no pollen or sodium selenite was added. 

#### 2.2.2. Total Phenolic Content 

Total phenolic content was determined after ultracentrifugation of the samples at 460,000 rcf (CP100NX Ultracentrifuge, Hitachi Koki, Tokyo, Japan). 

Total polyphenol content (TPC) was assessed by the Folin–Ciocalteu method [45,46]. Briefly, 90 μL of double-distilled water was added to 10 μL of sample/standard. After adding 10 μL of Folin–Ciocalteu reagent, the plate was shaken for 5 min. Afterwards, 100 μL of 7% Na_2_CO_3_ and 40 μL of double-distilled water were added. After incubating the microplate at room temperature for 60 min, absorbance values were recorded in triplicate at λ = 765 nm (CLARIOstar BMG Labtech, Ortenberg, Germany). The calibration curve was performed starting from a stock solution of 500 μg/mL gallic acid in 70% ethanol in the concentration range of 0–250 μg/mL.

To assess the total flavonoid content (TFC), 25 µL of sample/standard was mixed with 25 µL of 10% sodium acetate, 30 µL of 2.5% AlCl_3_, and 170 µL double-distilled water. After 45 min of incubation at room temperature, the absorbance was measured in triplicate at λ = 430 nm using a microplate reader. The calibration curve was performed starting from a stock solution of 500 μg/mL quercetin in 70% ethanol in the concentration range of 0–100 μg/mL [5,46]. 

In order to determine the total hydroxycinnamic acid content (HAT), 25 µL of sample/standard was mixed with 50 µL of 0.5 M HCl, 50 µL of 1% sodium nitrite, 1% sodium molybdate solution, 50 µL of 8.5% NaOH, and 75 µL of double-distilled water. Samples were shaken, and the absorbance values were recorded in triplicate at λ = 524 nm using a microplate reader. The calibration curve was prepared starting from a stock solution of 1 mg/mL chlorogenic acid in 70% ethanol in the concentration range of 0–300 µg/mL [46].

Total anthocyanin content (TAC) was determined by the pH differential method [46,47], by which 1 mL of sample was mixed with 1.5 mL of two different buffers: 0.025 M potassium chloride buffer, pH = 1, and 0.4 M sodium acetate buffer, pH = 4.5. After 30 min incubation at room temperature, the absorbance (A) was measured with a spectrophotometer (Ocean Optics UV-VIS-NIR DH-2000-BAL, Orlando, FL, USA) at 520 and 700 nm. The TAC was calculated according to the following formulas: $$∆A_{sample}=\left(A_{520}-A_{700}\right){pH}_{1.0}-\left(A_{520}-A_{700}\right){pH}_{4.5}$$
$$TAC\left(\frac{mg}{mL}\right)=(∆A\times MW\times DF)/(\varepsilon \times L),$$
where ΔA is the final absorbance of the samples, MW is the molecular weight of cyanidin-3-glucoside = 449.2 g/mol, DF is the dilution factor, Ε is the molar extinction coefficient of cyanidin-3-glucoside = 26,900 M^−1^ cm^−1^, and L is the optical path of the cuvette = 1 cm.

#### 2.2.3. Antioxidant Activity by DPPH, FRAP, and CUPRAC Assays

DPPH assay: The reaction mixture involved the addition of 100 μL of sample/standard to 100 μL of 0.3 mM DPPH solution prepared in absolute ethanol. After 30 min of incubation in the dark at room temperature and centrifugation at 6000 rcf, the absorbance was read in triplicate at λ = 517 nm using a plate reader. The calibration curve was performed in the concentration range of 0–0.15 mM, starting from a 1 mM Trolox stock solution prepared in 70% ethanol [46].

FRAP assay: We prepared 3 solutions, i.e., 300 mM sodium acetate buffer, pH = 3.6, 10 mM TPTZ in 40 mM HCl, and 20 mM FeCl_3_ in double-distilled water. The FRAP reagent was prepared by mixing 10 parts of 300 mM sodium acetate buffer solution, pH = 3.6, with 1 part of 10 mM TPTZ solution and 1 part of 20 mM FeCl_3_ solution (10:1:1). The FRAP reagent was incubated at 37 °C (MIR-154-PE, PHCBi Panasonic, Osaka, Japan). Over 15 µL sample/standard, 285 µL FRAP reagent was added. The samples were incubated at 37 °C in the same incubator in the dark for 30 min. After incubation, the samples were centrifuged at 6000 rcf to remove SeNPs and the absorbance was read in triplicate at λ = 593 nm using a microplate reader. The calibration curve was performed in the concentration range of 0–450 µM Trolox, starting from a stock solution of 10 mM Trolox in 70% ethanol [46].

CUPRAC assay: A total of 10 µL of sample/standard was mixed with 30 µL CuSO_4_ (5 mM), 30 µL neocuproine (3.75 mM), and 280 µL distilled water. After 30 min, the samples were centrifuged at 6000 rcf, and the absorbance was measured in triplicate at λ = 450 nm using a microplate reader. The calibration curve was performed starting from a stock solution of 10 mM Trolox in 70% ethanol in the concentration range of 0–2 mM [46,48].

#### 2.2.4. Soluble Silicon (Si) Content 

The determination of Si (mg/L) was carried out using a portable photometer (HI97705 Hanna Instruments, Smithfield, VA, USA), as per manufacturer’s instructions.

#### 2.2.5. Selenium Nanoparticles (Se^0^) Content

The total Se^0^ content was determined by the difference between the Se^0^ content from the supernatant obtained by centrifuging the samples at 3000 rcf and the supernatant obtained by ultracentrifuging the samples at 460,000 rcf (Universal 320R Centrifuge, Hettich, Tuttlingen, Germany). The calibration curve was performed starting from a stock solution of 0.1 M Na_2_SeO_3_ in the concentration range of 2–20 mM. An amount of 1 mL of sample/standard was mixed with 0.1 mL of 50 mM NH_2_OH. After N_2_ drying, the samples were resuspended in 0.2 mL 1 M Na_2_S. After mixing, the samples were incubated for one hour at room temperature. The absorbance was measured in triplicate at λ = 500 nm using a microplate reader, ClarioStar [49].

#### 2.2.6. Liquid Chromatography–Time-of-Flight/Mass Spectrometry Analysis (LC-TOF/MS) 

The samples were analyzed by LC-TOF/MS using an Agilent Technologies 1200 Infinity Series LC system (Santa Clara, CA, USA), consisting of 1260 quaternary pump, a 1260 ALS autosampler coupled to an electrospray ionization (ESI) source, and a 6224 TOF/MS, controlled by the Mass Hunter Acquisition software version B.06.01. The chromatographic separation was accomplished using an Agilent Poroshell 120 SB-C18 2.7 μm particle size, 4.6 mm × 50 mm (inner diameter × length) column, and as mobile phase, a mixture of 80% acetonitrile and 20% water containing 0.1% trifluoroacetic acid at a flow rate of 0.3 mL/min was used. For MS, the ESI was used in the positive mode with the following settings: dual spray needles for continuous infusion of reference mass solution, drying gas flowing at 3.0 L/min, nebulizer pressure of 15 psig, capillary voltage of 3500 V, and fragment or voltage of 175 V. The TOF was tuned and calibrated using Agilent ESI-TOF calibration and tuning mix. The data acquisition mass range was 100 to 1000 *m/z* at 10,002 transients/scan and 1 scan/s.

#### 2.2.7. High-Performance Liquid Chromatography (HPLC-DAD)

The chromatographic analysis of t-resveratrol from black tea was performed on a HPLC-DAD Agilent 1100 system with the method described by [50] with a flow of 1.2 mL/min using a Kromasil 100-5C18 5 μm particle size, 4.6 mm × 150 mm (inner diameter × length) column. A stock solution of 200.0 mg/L t-resveratrol was prepared in ethanol and the calibration standards in the range of 0.05–3.0 mg/L were obtained by appropriate dilution from stock solution. The peak areas of t-resveratrol versus concentrations were found to be linear (y = 74.135x − 2.221, R^2^ = 0.9996).

The chromatographic analysis of quercetin 3-rutinoside (Que-rut) from black tea was performed on a HPLC-DAD Agilent 1100 system with the method described by [51] on a Kromasil 100-5C18 column (150 mm × 4.6 mm, 5 μm). A stock solution of 156.0 mg/L quercetin 3-rutinoside was prepared in methanol and the calibration standards in the range of 0.16–31.2 mg/L were obtained by appropriate dilution from stock solution. The peak areas of Que-rut versus concentrations were found to be linear (y = 14.302x − 0.0346, R^2^ = 0.9999).

#### 2.2.8. Fourier-Transform Infrared Spectroscopy (FTIR)

An IRTracer-100 spectrometer (Shimadzu, Kyoto, Japan) was used for recording the FTIR spectra in Attenuated Total Reflectance (ATR) mode. The analyses were assessed as mean of 45 scans with a resolution of 4 cm^−1^ in the mid-IR spectral range of 7800–400 cm^−1^. The calibrations were performed based on the methodology proposed by the manufacturer. The exported files were slightly smoothed only in the wavenumbers range 4000–1501 cm^−1^ using the Savitzky–Golay method with 16 points of window and 2nd-degree polynomial order and further graphically overlaid using the OriginPro software version 9.9.5 from OriginLab Corporation (Northampton, MA, USA).

#### 2.2.9. Transmission Electron Microscopy–Energy-Dispersive X-ray (TEM-EDX) Analysis

Biogenic selenium nanoparticles were visualized by TEM using TECNAI F20 G2 TWIN Cryo-TEM (FEI) transmission electron microscope (Houston, TX, USA), and the elemental composition was assessed using the EDX detector (X-MaxN 80T—Oxford Instruments, Abingdon, Oxfordshire, UK). 

#### 2.2.10. Dynamic Light Scattering (DLS) and Zeta Potential Analysis 

DLS analysis and Zeta potential measurement were performed using AMERIGO™—Particle Size & Zeta potential Analyzer (Cordouan Technologies, Pessac, France), according to manufacturer’s instructions. The AmeriQ software version 3.2.3.0 was used for the result analysis. 

#### 2.2.11. D-Glucose/D-Fructose Content

D-glucose and D-fructose content was determined using the commercial Sucrose/D-Fructose/D-Glucose Assay Kit (Megazyme, Wicklow, Ireland), according to the kit protocol of the manufacturer.

#### 2.2.12. D-/L-Lactic Acid Content

D- and L-lactic acid content was determined using the commercial D-/L-Lactic Acid (D-/L-Lactate) Assay Kit (Megazyme, Wicklow, Ireland), according to the kit protocol.

### 2.3. Biocompatibility of SeNPs and Pollen-Fermented Kombucha Beverage

All biotests were performed on liophylised SeNPs and liophylised Kombucha beverages after centrifugation at 3000 rcf, without ultracentrifugation. The Kombucha beverages tested contained the SeNPs produced in situ.

#### 2.3.1. Cell Counting Kit-8 (CCK-8) Assay

Biocompatibility assays were performed on gingival fibroblasts (HGF-1, ATCC CRL-2014). The cells were maintained in DMEM (Dulbecco’s Minimum Essential Media) supplemented with 10% fetal bovine serum (FBS) at 37 °C under 5% CO_2_ atmosphere. The cells were seeded into 48-well plates at a density of 1 × 10^4^ cells/cm^2^. After 24 h in culture, the cells were treated with different concentrations of SeNPs: 0.1, 0.5, 2.5, 10 µg/mL, and SCOBY beverage in the absence and presence of pollen and selenite, respectively, K, and KPol5, KPol15, KPol25: 0.1, 1, 3, 5, 7 mg/mL, and maintained in the presence of suspensions for 24 h. For comparative purposes, negative (untreated cells, C−) and positive (cells treated with 7.5% DMSO, C+) cytotoxicity controls were performed in parallel. For the CCK-8 assay, after suspensions removal, the cells were treated with CCK-8 prepared in DMEM supplemented with FBS (the ratio between CCK-8 and culture medium being 1:10). After 2 h of incubation at 37 °C, in 5% CO_2_, the plates were gently shaken, and the liquid was transferred to 96-well plates for absorbance reading at 450 nm using a microplate reader. The assay was performed in triplicate.

#### 2.3.2. LIVE/DEAD Assay

For the LIVE/DEAD assay, the HGF-1 cells were seeded, incubated, and treated as previously described in Section 2.3.1. After suspension removal, the cells were washed once with DMEM and incubated with calcein AM/EthD-1 (2 M:4 µM) for 10 min. Afterwards, the cells were washed again with DMEM and then visualized with CELENA X High Content Imaging System (Logos Biosystems, Gyeonggi-do, South Korea,). The image acquisition was carried out using CELENA X Explorer software version 1.0.5 (4× objective). The image analysis was performed with CELENA X Cell Analyzer version 1.5.2. 

#### 2.3.3. Assessment of Cell Morphology

The HGF-1 cells were maintained in standard culture conditions (untreated cells) and subjected to 24 h treatment as previously described in Section 2.3.1, with the concentrations that induced the highest increase in the number of metabolically active cells, i.e., 2.5 µg/mL SeNPs, 3 mg/mL K, 3 mg/mL KPol5, 5 mg/mL KPol15, and 7 mg/mL KPol25. The cell morphological features were revealed on the gingival fibroblasts by fluorescent labeling of actin cytoskeleton with Alexa Fluor 488-coupled phalloidin. In addition, the nuclei were marked with DAPI (4′,6-diamidino-2-phenylindole). After suspension removal, the cells were washed twice with DMEM, once with PBS, and then fixed with 4% paraformaldehyde prepared in phosphate-buffered saline (PBS) for 20 min. Subsequently, they were washed with PBS three times and permeabilized with 0.1% Triton X-100/2% bovine serum albumin (BSA) for 30 min. After three more washes with PBS, the cells were incubated for 15 min with Alexa Fluor 488-coupled phalloidin for cytoskeleton labeling. Afterwards, cells were washed with PBS three times and incubated with DAPI for 10 min in order to stain the nuclei. After three more washes with PBS, cells were analyzed with CELENA X High Content Imaging System. The image acquisition was performed using CELENA X Explorer software version 1.0.5 (10× objective). The image analysis was performed with CELENA X Cell Analyzer version 1.5.2. 

#### 2.3.4. In Vitro Antioxidant Activity of SeNPs and Pollen-Fermented Kombucha Beverage

HGF-1 cells were seeded and cultured for 24 h, as shown in Section 2.3.1. Subsequently, they were treated with different concentrations of products in the presence of 37 µM hydrogen peroxide solution (reactive oxygen species (ROS) inducer) and maintained in the presence of suspensions for 24 h. In parallel with these samples, a negative control (untreated cells) and a positive control (cells treated with 37 mM H_2_O_2_ and without the tested products) were investigated. After 24 h, the cells were washed once with DMEM and incubated for 30 min at 37 °C, under 5% CO_2_ with 10 µM 2’,7’-dichlorodihydrofluorescein diacetate (H_2_DCFDA) solution prepared in DMEM from a stock solution of 10 mM H_2_DCFDA in DMSO. Following the dye removal, cells were washed twice with DMEM and once with PBS. After image acquisition and analysis (4× objective) using CELENA X Explorer software version 1.0.5, respectively CELENA X Cell Analyzer version 1.5.2., PBS was removed, and cells were treated with RIPA lysis buffer. Plates were incubated for 5 min on ice, afterwards the cell lysates were aspirated into Eppendorf tubes and centrifuged at 19,000 rcf for 10 min. The fluorescence intensity was measured using a microplate reader (485 nm excitation, 530 nm emission) [52]. The assay was performed in triplicate.

### 2.4. Pollen Characterization

#### 2.4.1. Extraction of Phenolic Compounds from Fresh Polyfloral Pollen 

Four methods of extracting phenolic compounds from fresh polyfloral pollen were tested:
(a)A total of 0.5 g of fresh polyfloral pollen was mixed with 10 mL of 80% methanol [53]. The mixture was kept for 1 h in an ultrasonic bath at 30 °C. The methanolic extract was centrifuged for 30 min at 4 °C, 7500 rcf. After N_2_ drying, the sample was resuspended in 70% ethanol. (b)A total of 0.5 g of fresh polyfloral pollen was mixed with 10 mL of 50% ethanol [53]. The mixture was kept for 1 h in an ultrasonic bath at 30 °C. The ethanolic extract was centrifuged for 30 min at 4 °C, 7500 rcf. After N_2_ drying, the sample was resuspended in 70% ethanol. (c)A total of 1 g of fresh polyfloral pollen was mixed with 0.5 mL of 0.5% hexamethyl tetramine (HMTA), 10 mL of acetone, and 1 mL of 0.1 N HCl. The mixture was kept for 30 min at 100 °C with reflux. The extract was centrifuged for 30 min at 4 °C, 7500 rcf. Afterwards, we mixed 10 mL of the supernatant with 20 mL of double-distilled water and 25 mL of ethyl acetate in a separation funnel. Liquid–liquid extraction was performed three times to increase the extraction yield [54]. After N_2_ drying, the sample was resuspended in 70% ethanol. (d)A total of 0.5 of fresh polyfloral pollen was mixed with 10 mL of 2 M NaOH, and the mixture was kept for 1 h in an ultrasonic bath at 30 °C. Afterwards, the pH of the sample was decreased with 37% HCl at 1.8. The extract was centrifuged for 30 min at 4 °C, 7500 rcf. The supernatant was mixed with ethyl acetate in a separation funnel using a ratio of 1:1. Liquid–liquid extraction was performed three times to increase the extraction yield. After N_2_ drying, the sample was resuspended in 70% ethanol.

The pollen extracts were analyzed by LC-MS, as described in Section 2.2.9.

#### 2.4.2. Scanning Electron Microscopy (SEM)–Energy-Dispersive X-ray (EDX) Analysis of Pollen

SEM micrographs of pollen were acquired with TM4000Plus II tabletop electron microscope (Hitachi, Tokyo, Japan) working with 15 kV electron acceleration voltage, secondary electrons (SE) detector, standard (M) vacuum mode, and 100× magnification. The elemental composition was assessed using an EDS detector AZtecLiveLite (Oxford Instruments, Abingdon, UK). The adjustments were conducted based on the manufacturer’s proposed methodology.

#### 2.4.3. Fourier-Transform Infrared Spectroscopy (FTIR)

FTIR analysis of fresh polyfloral pollen and fermented pollen was performed as described in Section 2.2.6. 

### 2.5. Modulation Effects of Kombucha Fermentation on Bacterial (Nano)Cellulose (BNC) 

#### 2.5.1. Bacterial Cellulose (BC) Weight 

The never-dried bacterial cellulose membrane was weighed in order to determine the influence of Kombucha modulation.

#### 2.5.2. Bacterial Nanocellulose (BNC) Production

BC went through several processes in order to obtain BNC:
(a)Membrane bleaching process: A total of 300 g of initial bacterial cellulose membranes was immersed in 600 mL of 1 M NaOH for 2 h at 60–80 °C using an ultrasonic bath Elmasonic P (580 W, 37 Hz, Elma Schmidbauer GmbH, Singen, Germany) [55]. Subsequently, 600 mL of 5% hydrogen peroxide solution was added, and the membranes were kept in the ultrasonic bath for another 6 h. After bleaching, the membranes were washed with double-distilled water until neutral pH. (b)Grounding process (milling): The washed membranes were subjected to the grounding process for 1 h using a blender.(c)Microfluidization: A suspension of 1% ground membranes in double-distilled water was prepared. The suspension was microfluidized using LM20 microfluidizer (Microfluidics, Worcester, UK). The samples were collected after 1, 10, and 20 cycles/passes.

The morphology and structural features of BC and BNC were investigated by FTIR and SEM as previously described (Section 2.2.6 and Section 2.4.2, respectively).

#### 2.5.3. X-ray Diffraction Analysis (XRD)

X-ray diffraction analysis was performed using a Rigaku-SmartLab diffractometer (Rigaku Corporation, Tokyo, Japan) working at 45 kV voltage and 200 mA current intensity, with a Cu_Kα1_ incident radiation at 1.54059 Å wavelength in the 2θ range 5–50°, with 0.02° resolution and 4°/min scanning speed. The diffraction spectra were smoothed in the PDXL software version 2.7.2.0 using the B-Spline model for Chi = 1, followed by deconvolution, background subtraction, peak identification, and calculation of the crystallinity degree (Xc, %) as the ratio between the area of crystalline peaks over the total peaks area. 

### 2.6. Statistical Analysis 

IBM SPSS Statistics software version 26.0.0.0 was used for statistical analysis (One-Way ANOVA). 

## 3. Results

### 3.1. RSM Analysis of Pollen-Fermented Kombucha Beverage with In Situ Biosynthesized SeNPs

Table 2 presents the *p*-value after the ANOVA analysis, which shows the statistical significance of the resulting model of the three factors (SCOBY, pollen, and sodium selenite) and the interaction between all factors. All the models obtained are statistically significant (*p* value < 0.05). The R^2^ and Adjusted R^2^ values are shown in Appendix A, as well as the coded equations (Appendix A). 

All values obtained for each response are presented in Appendix A. The analysis of the response surfaces (Figure 1) indicated that there was a decrease in the total content of polyphenols, flavonoids, hydroxycinnamic acids, and antioxidant activity (AOA) measured by DPPH, FRAP, and CUPRAC assays by increasing the amount of sodium selenite (Figure 1a–f) following the ultracentrifugation of the samples. The soluble Si content increased by increasing the amount of pollen due to the release of the pollen biosilica-rich content during the fermentation process. The concentration of SCOBY increased and decreased the soluble Si at the highest and lowest selenite concentrations, respectively. Selenite increased and decreased the soluble Si at the lowest and the highest SCOBY added, respectively (Figure 1g). The Se^0^ content and Se^0^ biosynthesis yield were influenced by the amount of sodium selenite. The minimum concentrations of sodium selenite (10 mg) from the experimental design resulted in very low Se reduction to Se^0^ forms, and the highest selenite concentration (370 mg) determined only 50–60% reduction, regardless of the other variables (Figure 1h,i). The quantity of SCOBY did not have a significant effect on Se^0^ content and Se^0^ biosynthesis yield.

The Pearson correlation is useful in establishing the correlations between different responses. The Pearson correlation plot (Figure 2) gives an overall picture about the correlations, i.e., the more ordered the plot becomes (the points line up in an orderly pattern), the more there is a significant direct proportionality or inverse proportionality relationship between the responses.

Appendix A helps us in correlating all the responses and also in understanding how significant the relationship between the responses is. Between TPC and TFC, HAT, DPPH, FRAP, and CUPRAC there is a directly proportional relationship, i.e., when one response increases, the other increases. Apart from FRAP where the correlation is significant at the 0.05 level, for all other responses, the correlation is significant at the 0.01 level. Between TPC and Se^0^ content or Se^0^ biosynthesis yield, there is an inverse proportionality relationship at the 0.01 level. The TFC content is in a direct proportional relationship at the 0.01 level with TPC, HAT, DPPH, FRAP, and CUPRAC and in an inversely proportional relationship with Se^0^ content and Se^0^ biosynthesis yield. In the case of HAT, there is no significant relationship with FRAP, but there is a significant relationship at the 0.01 level with TPC, TFC, DPPH, and CUPRAC of direct proportionality and inverse proportionality with Se^0^ content and Se^0^ biosynthesis yield. DPPH is in a direct proportional relationship with TPC, TFC, HAT, FRAP, and CUPRAC at the 0.01 significance level and in an inverse proportional relationship at the 0.05 level with Se^0^ content. In the case of FRAP, there is only a direct proportional relationship with TFC and DPPH at the 0.01 level in terms of statistical significance and at the 0.05 level with TPC. In the case of CUPRAC, the direct proportionality relationship is established with TPC, TFC, HAT, and DPPH at the 0.01 level, and the inverse proportionality relationship is established with Se^0^ content at the 0.01 level and with Se^0^ biosynthesis yield at the 0.05 level. There is no relationship between the soluble silicon content and the other responses. Between absolute Se^0^ content and TPC, TFC, HAT, and CUPRAC, there is an inverse proportionality relationship at the 0.01 level, and with DPPH the significance is at the 0.05 level. The direct proportionality relationship between Se^0^ content is established with the Se^0^ biosynthesis yield at the 0.01 level. Between Se^0^ biosynthesis yield and TPC, TFC, and HAT, there is a directly proportional relationship at the 0.01 level, and the relationship with CUPRAC is significant at the 0.05 level.

The optimal model represented by the desirability function was generated based on the maximization of Se^0^ biosynthesis yield (Figure 3a) and on the maximization of both Se^0^ biosynthesis yield and absolute Se^0^ content (Figure 3b). As mentioned above, at low concentrations of sodium selenite, almost no SeNPs are formed. Figure 3a indicates that the maximum yield of selenium biotransformation is reached at sodium selenite concentrations of about 4 mM (207 mg selenite—0.69 mg/mL). In the case of generating desirability depending on both the content of Se^0^ and the biosynthesis yield of Se^0^, it can be seen in Figure 3b that the optimal model indicates sodium selenite concentrations of approximately 7 mM (363 mg selenite—1.21 mg/mL).

The desirability to obtain the maximum yield (maximum transformation of selenite into SeNPs) prevails over obtaining the maximum absolute values of SeNPs, which in too high concentrations could become toxic. 

Since the biosynthesis yield of SeNPs was almost 100% in the case of the experimental variant with 30 mL SCOBY, 25 g of pollen, and 190 mg of selenite (KPol25), this sample was chosen as the optimal model from the designed set. For the determination of all physicochemical and biological properties, the sample was accompanied by the experimental variants with 30 mL SCOBY, 5 g of pollen, and 190 mg of selenite (KPol5), with 30 mL SCOBY, 15 g of pollen, and 190 mg of selenite (KPol15), as well as by the Kombucha beverage in which no pollen or selenite was added (K). In the case of some analyses, the variant with 30 mL SCOBY, 15 g pollen, and 10 mg selenite (KPol) was also included for comparison.

### 3.2. Characterisation of Selected Kombucha–Pollen Beverage Variants and Biosynthesized SeNPs

Increasing pollen concentration leads to increasing soluble silicon concentration, from 4.25 ± 0.5 mg/L Si in the sample containing 5 g of pollen—KPol5—to 5.8 ± 0.5 mg/L in the sample with 15 g of pollen—KPol15—and reaches 11.5 ± 1.5 mg/L in the sample with 25 g of pollen—KPol25. Sample K contains only 3.5 ± 0.5 mg/L soluble silicon (Figure 4).

The compounds from Kombucha beverages identified by LC-TOF/MS analysis are presented in Table 3. These included polyphenols, fatty acids, two-amino acid (γ-Aminobutyric acid, known as GABA and valine), and one polyamine (spermidine). Only caffeine was present in all tea beverages. Other compounds were present only in some samples or even only one sample (linolenic acid and the last three compounds).

TPC analysis showed a content of polyphenols in black tea of 86.1 ± 3.1 µg/mL. From Figure 5a, it can be seen that the total polyphenol content is reduced by about half in sample K to 40.4 ± 3.7 µg/mL, followed by further decreases in sample KPol (26.5 ± 1 µg/mL), KPol5 (18.1 ± 0.8 µg/mL), and KPol15 (11.6 ± 0.9 µg/mL), after which an increase is observed in sample KPol25 up to 16.6 ± 1.5 µg/mL. The black tea has a total flavonoid content of 3.6 ± 0.05 µg/mL. There is only a significant decrease in TFC in KPol15 (2.2 ± 0.2 µg/mL); the rest of the samples, respectively, K, KPol, KPol5, and KPol25 have similar TFC content to black tea, with the differences between them being not statistically significant (4.2 ± 0.07 µg/mL for K, 4.8 ± 0.3 µg/mL for KPol, 4.3 ± 0.2 µg/mL for KPol5, and 3.6 ± 0.07 µg/mL for KPol25). Black tea has a HAT content of 29.4 ± 1.4 µg/mL. The HAT trend is similar to that of TPC, i.e., the total hydroxycinnamic acid content is 35.5 ± 0.4 µg/mL in sample K, after which there is a significant decrease in HAT in sample KPol to 22.1 ± 2.1 µg/mL. The HAT content further decreases significantly in samples KPol5 to 14.4 ± 1. 5 µg/mL and KPol15 to 4.8 ± 0.3 µg/mL, with a marginally significant increase compared with KPol15 is observed in the sample KPol25 to 9.6 ± 0.9 µg/mL (Figure 5a). The TAC content of black tea was 0.25 ± 0.06 µg cyanidin-3-glucoside/mL. 

Regarding the antioxidant activity measured by the DPPH method, there are no significant differences between samples (Figure 5b). By the FRAP method, there are only small variations between samples K, KPol, KPol5, and KPol25, but they are not statistically significant; the only significant difference being the decrease in the antioxidant activity in the case of sample KPol15 to 1295.3 ± 75.4 µM Trolox equivalents (almost 3× reduction). The CUPRAC method showed the highest antioxidant activity for KPol (1504.6 ± 26.8 µM), followed closely by K (1487.1 ± 152.7 µM) and KPol5 (1478.6 ± 164.8 µM), with only marginally significant differences between these samples. The antioxidant activity decreases for KPol15 (1081.1 ± 44.3 µM), and there is a slight increase for KPol25 (1131.4 ± 9.31 µM) compared with KPol15 (Figure 5b).

The selected SCOBY-Kombucha beverages were freeze-dried and spectroscopically compared with the control and with the SeNPs based on FTIR analysis. Figure 6 evidences the hydrogen bonds of OH and NH_2_ in the diagnostic region (4000–2800 cm^−1^) and, in the fingerprint region, the following bands are marked: the specific amide bands of proteins and amino acids, the absorption bands of structural carbohydrates STCHO and the bands of (poly)saccharides. SCOBY Kombucha as control contains proteins/amino acids, (poly)saccharides, lipids and STCHO carbohydrates. The addition of 15 g pollen in the first sample KPol compared with the control K is mainly increasing the protein bands with the amide I peak around 1703 cm^−1^ and the amide II band around 1638 cm^−1^. The sample KPol5 with 5 g pollen and 190 mg Na_2_SeO_3_ has weaker bands and a more compact saccharides band and two additional peaks around 800 cm^−1^ that will be later correlated with possible Se bonds. The obtained SeNPs seem to receive by fermentation a corona of amino acids with peaks at 1722, 1638, and 1541 cm^−1^, and the region 900–800 cm^−1^ suggests particular Se bonds that will be further investigated.

The ATR-FTIR analysis reveals in Figure 7a the absorption bands specific to the Se-O and Se=O bonds in the initial sodium selenite Na_2_SeO_3_, respectively, as well as the disappearance of the main band at 714 cm^−1^ characteristic for the selenite ion SeO_3_^2−^ after the biosynthesis experiments with Kombucha. The absorption bands at 874 cm^−1^ and 777 cm^−1^ suggest the presence of Se-O and Se=O bonds, respectively, of SeNPs, while the bands at 1541, 932 and 891 cm^−1^ suggest the formation of novel bonds of Se with C, N, and O from proteins, polyphenols, and/or oligosaccharides, generically referred to as “biocorona”. 

The “biocorona” was further investigated by spectroscopic comparison with gallic acid as a representative compound from the hydroxycinnamic acids class, with catechin hydrate as a representative flavonoid, respectively, with BSA as a representative protein (Figure 7b), as well as with other compounds similar to those detected by LC-MS (Appendix A). It is observed that the IR spectrum of SeNPs show similarities to proteins, flavonoids, and polyphenolic acids. Characteristic bands of proteins are the amide I, II, and III bands, respectively, amide I at 1638 cm^−1^ specific to C=O and C-OH bond vibrations, amide II at 1541 cm^−1^ specific to C-N vibration, respectively, and amide III around 1450–1300 cm^−1^ specific to C-N-H vibrations. Another similarity between SeNPs and BSA is the band at 1076–1082 cm^−1^, characteristic of C-N bonding in amino acids, amines, and amides. The presence of the aromatic Ar-OH band at 1223 cm^−1^ in SeNPs, around 1200 cm^−1^ in gallic acid and at 1242 cm^−1^ in catechin and BSA, can be observed. Common peaks with representative compounds are seen also in the region characteristic of intermolecular H-bonds, 3200–3300 cm^−1^ (Figure 7b and Appendix A). The “biocorona” of SeNPs depends on the biosynthesis method; therefore, various spectra may be obtained [56,57].

TEM analysis showed quasi-spherical SeNPs in the range of 20 to 100 nm (Figure 8a). EDX analysis revealed that the main elements are C, O, and Se (Figure 8b). 

We fitted the DLS autocorrelation function based on the three models provided by the software: Cumulants, Pade Laplace, and SBL. The Cumulants function indicated a Z-average value of 388.78 nm with a polydispersity index (PDI) of 0.2899, which indicates a rather polydisperse distribution. The simulation based on number indicated a mean value of 192 nm, representing the most abundant NP size. This model did not ideally fit the experimental data, the residues exceeding the accepted limit (Appendix A). The Pade Laplace indicated NP of 30 nm as the majority population, but we excluded this model because of the PDI suggested value and the data from TEM, which indicated mostly NP core > 50 nm. The mean values from intensity and volume were similar to the values from the Cumulant model. The SBL model indicated a monomodal distribution with a mean diameter of 431 nm based on intensity (Appendix A). When shifting to volume (Appendix A), a bimodal distribution was generated with mean diameters shifted to lower values, 51 nm (34% volume) and 136 nm (66% volume). The number distribution showed only the small NP with the mean diameter 49 nm (Appendix A and Table 4). The autocorrelation function and its simulations are shown in Appendix A.

The SeNPs showed an electronegative Zeta potential of −8.8 ± 0.35 mV, which indicates a slight tendency towards aggregation (Appendix A). 

Following the determination of antioxidant activity using the three different methods, i.e., DPPH, FRAP, and CUPRAC, and three different concentrations of SeNPs, i.e., 66, 133, and 200 μg/mL, it can be observed in Figure 9 that the antioxidant activity increased by increasing the SeNPs concentration. By the DPPH method, the antioxidant activity of SeNPs increased from 46.3 mM ± 3.1 TE at 66 µg/mL SeNPs concentration to 85.7 ± 0.6 mM TE at 200 µg/mL concentration. In the case of the FRAP method, the antioxidant activity varied according to the SeNPs concentration between 104.4 ± 2.3 mM TE and 155.28 ± 3.09 mM TE. In the case of CUPRAC method, the lowest antioxidant activity was 94.2 mM TE at 66 µg/mL SeNPs concentration and reached 179.8 ± 6.4 mM TE at the highest concentration of SeNPs.

The dynamics of yeast and (fructophilic) lactic bacteria populations were analyzed by measuring the D-glucose, D-fructose, and D- and L-lactic acid content (Figure 10). For D-glucose content, there was a significant increase (9.6 ± 0.05 g/L) in the case of the KPol5 sample in comparison with control K (6.5 ± 0.3 g/L), after which there was a slight decrease in the case of the KPol15 sample (9.1 ± 0.5 g/L), but it was not statistically significant, as well as a marginally significant decrease in the case of the KPol25 sample (8 ± 0.009 g/L). In terms of D-fructose content, the K control has approximately 1.9 ± 0.2 g/L. In sample KPol25, there is a marginally significant decrease in D-fructose content (1.3 ± 0.01 g/L), followed by a significant decrease of about 40% in the case of KPol15 (1.1. ± 0.06 g/L), until the total reduction in D-fructose content in sample KPol5 (Figure 10a).

In the case of D-lactic acid content, there is a marginally significant increase in KPol5 (0.3 ± 0.0009 g/L) followed by significant increase in KPol15 (0.5 ± 0.03 g/L) and KPol25 (1.1. ± 0.1 g/L) compared with the K control (0.08 ± 0.006 g/L). The L-lactic acid content increases significantly in all samples, i.e.,0.8 ± 0.05 g/L in KPol5, 1.1 ± 0.05 in KPol15, and 1.9 ± 0.04 g/L in KPol25 in comparison with K control (0.153 ± 0.010 g/L). The amount of pollen was positively correlated with both the D- and L-lactic acid concentrations (Figure 10b).

The CCK-8 assay results shown in Figure 11a after cell exposure to various concentrations of SeNPs indicated a marginally significant increase in the number of metabolically active cells at the lowest tested concentrations, i.e., 106 ± 0.4% at 0.1 µg/mL SeNPs and 108.4 ± 0.6% at 0.5 µg/mL. The maximum potential of inducing cell proliferation was recorded at 2.5 µg/mL SeNPs, i.e., 111.3 ± 2.7%, which was statistically significant. Note that the CCK-8 assay results can be correlated with the fluorescence microscopy images presented in Figure 11b.

The cell morphology analysis revealed that the cytoskeleton (green fluorescence) was well organized in a fibrillar structure with abundant actin filaments that improve the mechanical stability of the nucleus (blue fluorescence), as shown in Figure 12. 

For the in vitro antioxidant activity, we chose the concentrations of SeNPs that exhibited the highest degree of inducing an increase in the number of metabolically active cells, i.e., 0.5 µg/mL and 2.5 µg/mL. There was a reduction of about 70–80% in the amount of reactive oxygen species with respect to C+ (160.4 ± 4% of C−) after 24 h following the SeNPs treatment in the presence of ROS inducer (57.5 ± 2.8% of C− for 0.5 µg/mL SeNPs and 43.7 ± 1.3% of C− for 2.5 µg/mL SeNPs). Figure 13a can be correlated with the fluorescence microscopy images from Figure 13b acquired after labeling total ROS with H_2_DCFDA.

By assessing the effects of the pollen-fermented Kombucha beverage on cell viability and proliferation using the CCK-8 assay, it was found that there was a dose-dependent increase in the number of metabolically active cells in the K sample, starting from 104.1 ± 1.7% for 0.1 mg/mL K and reaching 107.4 ± 1.8 for 1 mg/mL K, but these increases are not statistically significant (Figure 14a). The maximum significant potential for inducing cell proliferation is reached at 3 mg/mL K (114 ± 2.4%), after which there is a decrease in cell viability to 108.5 ± 1.24% for 5 mg/mL K with a further significant decrease in comparison with C− at 7 mg/mL K (72.1 ± 1.9%). The sample KPol5 was biocompatible at all tested concentrations, without significant changes compared with C− (cell viability ranging between 102.7 ± 0.7% for 0.1 mg/mL KPol5 and 98.4 ± 4.1% for 7 mg/mL KPol5). Regarding the KPol15 sample, with the exception of the concentration of 0.1 mg/mL where cell viability and proliferation were not statistically significant (106.1 ± 1.9%), in the case of all other concentrations, there was a significant increase in the number of metabolically active cells: 113.1 ± 3.2% for 1 mg/mL KPol15 and 114.4. ± 4.05% for 3 mg/mL KPol15, with a maximum potential for inducing cell proliferation at 5 mg/mL KPol15 (119.4 ± 0.8%). At 7 mg/mL KPol15, the cell viability slightly decreased, but it stayed above the negative cytotoxicity control (115.7 ± 0.5%). In the sample with 25 g of pollen and where the SeNPs biosynthesis yield was over 95% (KPol25), cell viability and proliferation increased with increasing the KPol25 concentration from 102.6 ± 2.1% for 0.1 mg/mL KPol25 to 105.6 ± 1.7% for 1 mg/mL KPol25 and 108.3 ± 0.9% for 3 mg/mL KPol25, these increases being not statistically significant. The increase became statistically significant at 5 mg/mL KPol25 treatment, i.e., 117.2 ± 3.3%, and the maximum potential for inducing cell proliferation was recorded at 7 mg/mL KPol25 concentration (123.7 ± 0.4%). The CCK-8 assay results can be correlated with the fluorescence microscopy images presented in Figure 14b–m and Appendix A.

There were no changes in cell morphology following cells treatment with the Kombucha beverages, the cytoskeleton being well organized with abundant actin filaments (Appendix A).

For the in vitro antioxidant activity, we chose the highest concentrations of K, KPol5, KPol15, and KPol25, respectively, 3, 5, and 7 mg/mL. The in vitro antioxidant activity shown in Figure 15a highlighted a significant decrease in ROS generation in all samples. The highest ROS reduction capacity in the case of sample K was observed at the lowest concentrations (26.6 ± 1.9% ROS of C− and 16.7% of C+ for 3 mg/mL K, 27.4 ± 0.6% ROS of C− and 17.1% of C+ for 5 mg/mL K). At the highest concentration, 7 mg/mL, there was a reduction in the antioxidant effect (40.08 ± 1.3%) compared with control sample or a pro-oxidant effect as compared with 3 and 5 mg/mL K. A similar trend can be observed in the case of the KPol5 sample, but its ability to decrease the total ROS at 3 and 5 mg/mL concentrations is lower in comparison with K (38.3 ± 1.1%, 39.2 ± 1.1.% and 40.1 ± 3.7% ROS of C− for, respectively, 3, 5, and, 7 mg/mL KPol5). 

In the case of KPol15, which contained 15 g of pollen and where the biosynthesis yield of SeNPs was about 50%, the antioxidant activity increased by increasing the concentration of KPol15 (45.4 ± 1.4% ROS, 43.2 ± 2.7%, and 39.02 ± 2.1% ROS of C− for 3, 5, and 7 mg/mL, respectively). The ROS scavenging capacity of the KPol25 sample follows the same trend as the KPol15 sample, but the difference between the two samples becomes statistically significant at higher KPol25 concentrations. The highest KPol25 concentration tested, 7 mg/mL, shows the highest antioxidant activity of all samples. The values are 36.8 ± 0.4%, 30.4 ± 0.2, and 25.2 ± 1.5% ROS of C− for 3, 5, and 7 mg/mL KPol25, respectively). Figure 15a can be correlated with the fluorescence microscopy images from Figure 15b–i and Appendix A acquired after labeling total ROS with H_2_DCFDA.

### 3.3. Physicochemical Properties of Pollen Fermented with a Kombucha Consortium and Analysis of Phenolic Content

SEM-EDX analysis of fresh polyfloral pollen and fermented pollen showed that pollen fermentation with Kombucha consortium produced a product morphologically similar to bee bread (Figure 16, Appendix A). The fermented pollen denoted PK25 (from sample KPol25) seemed to be enriched with Se, most probably in the form of SeNPs, based on the EDX analysis.

The effects of pollen fermentation with SCOBY and Na_2_SeO_3_ on the pollen spectra are depicted in Figure 17. The diagnostic region between 3700–3000 cm^−1^ evidences a high number of hydrogen bonds in crude and lyophilized pollen due to -OH and -NH_2_ functional groups of polyphenols, flavonoids, amino acids, and proteins [58,59]. This wide band has a correspondence for the same functional groups at half wavenumbers, respectively, 1750–1500 cm^−1^, where the amide bands can be differentiated in amide I for C=O and C-OH vibrations, amide II for C-N vibrations and amide III for C-N-H vibrations, respectively. The region 2950–2850 cm^−1^ corresponds to C-H vibrations and is usually sharper for aliphatic chains like in lipids [58]. This region also has a correspondence in the fingerprint region at half wavenumbers for the asymmetric and symmetric stretching of C-C-H chains in lipid chains and structural carbohydrates (STCHO) [60], while in the case of pollen, these absorption bands can also be assigned to aliphatic α-pyrone chains in sporopollenin, the toughest natural polymer, therefore surnamed “the diamond of the plant world” [61,62].

Fermented pollen, especially PK25, loses a part of the hydrogen bonds and especially -OH groups around 3400 cm^−1^, while the amide I and II bands are better differentiated, suggesting also a fragmentation of the initial hydrogen-bonded structure. The saccharides region 1200–900 cm^−1^ is not drastically affected by fermentation at low selenite concentration (10 mg) in PK, while at higher selenite concentration of 190 mg for PK25, the saccharides band splits in two bands assigned to C-N in amino acids around 1082 cm^−1^ and to C-O in oligosaccharides around 1022 cm^−1^. The PK25 sample contains bands characteristic to the IR spectrum of SeNPs, suggesting that PK25 is fermented pollen enriched by SeNPs, which correlates with the SEM-EDX data (Figure 17).

### 3.4. Modulation Effects of Kombucha Fermentation on Bacterial (Nano)Cellulose (BNC)

An increase in bacterial cellulose weight was observed for BCKPol5 (80.9 ± 11.5%), BCKPol15 (94.2 ± 3.1%), and BCKPol25 (135.5 ± 23.4%) compared with the control BCK (25.9 ± 1.0%) (Figure 18a). The appearance of the BCK and BCKPol25 bacterial cellulosic membranes can be seen in Figure 18b. The presence of pollen and selenite induced a whiter cellulose membrane compared with the Kombucha control.

Topographical analysis of the initial bacterial cellulose (the membrane) obtained from the control Kombucha sample (BCK) indicated a rather disordered structure (Figure 19a,b). Following the modulation of Kombucha fermentation, it can be seen that there is a gradual tendency of cellulose fibers to rearrange into a more ordered structure in the presence of pollen and selenite (Figure 19). The fermentations in the presence of 15 or 25 g pollen showed pollen grains entrapped in the cellulose membrane (Figure 19). The sample BCKpol25 (190 mg selenite and 25 g pollen) had traces of Se detected in the membrane as detected by EDX analysis, with BCKpol25 being the sample with the highest yield of sodium selenite biotransformation (Appendix A).

The XRD analysis of the initial Kombucha membranes was further compared in Figure 20 with the ones purified (BCKw) and microfluidized by 1, 10, and 20 passes. All the samples contain the Iα (PDF card No. 00-056-1719) and Iβ (PDF card No. 00-060-1502) cellulose allomorphs together with amorphous cellulose (PDF card No. 00-060-1501) in different amounts suggested both by the crystallinity index and the relative intensities. The membrane with the highest pollen amount (BCKPol25) shows an initial crystallinity higher than the control BCK, 61% compared with 56%. The amorphous peak marked at 18.9° in BCKPol25 has a lower relative intensity and a different shape in comparison with the amorphous peak of the control placed at 19.4°, suggesting a different composition. By purification, the crystallinity increases significantly to 94 and 96% for BCKPol25 and control (without pollen and selenite), respectively, due to the removal of the interfibrillar amorphous melanoidins [63]. By further processing through microfluidization, in order to obtain nanocellulose, it was observed that after 1 pass for BCKPol25, respectively, 10 passes for the control, the crystallinity slightly increases further, mainly due to the arrangement of the cellulose microfibrils. After 20 passes, the crystallinity of the control decreases to 75%, suggesting the braking of the microfibrils into nanofibrils and amorphous cellulose, while the crystallinity of BCKPol25 decreases only to 92%, suggesting a stronger fibrillar structure. This stronger structure might be explained by the higher ratio between the relative intensities of Iβ peak compared with the Iα and Iαβ crystal structures in BCKPol25 compared with the control. This might suggest that the Na_2_SeO_3_ influenced the bacteria in creating a stronger cellulose structure, possibly due to the known toxic effect of selenite at high concentrations. The samples BCK and BNC of KPol5 and KPol15 were similarly analyzed and presented in Appendix A. The main observation was that the intermediary pollen concentrations of 5 g in BCKPol5 and 15 g in BCKPol15 showed intermediary characteristics of bacterial cellulose based on the relative intensities of Iαβ (14.6°), Iα (16.9°), and Iβ (22.68°) cellulose allomorphs. The Iαβ relative intensity was 0.82 for BCK, 0.41 for BCKPol5, 0.29 in BCKPol15, and 0.27 in BCKPol25 (Figure 20 and Appendix A). The Iα relative intensity was 0.74 for BCK, 0.56 in BCKPol5, 0.41 in BCKPol15, and 0.40 in BCKPol25. The decrease in Iα and Iαβ relative intensities correlates with the increase in the stronger Iβ crystalline arrangement as a two-chains monoclinic structure. The cellulose crystallinity behaved similarly to the crystallinity of cellulose from KPol5 and KPol15.

For a better overview of the SCOBY fermentation process in the presence of pollen and Na_2_SeO_3_, the FTIR spectra of the bacterial cellulose membranes formed at the air–liquid interface in the experimental conditions with 190 mg Na_2_SeO_3_ and 5, 15, and 25 g of pollen were compared in Figure 21 with the control membrane without pollen (BCK) and with the corresponding bacterial nanocelluloses obtained by purification and microfluidization by 20 passes. Additionally, in Figure 21, the SeNPs spectrum is presented in order to investigate the possible presence of SeNPs in the initial membranes.

From Figure 21 and Appendix A, the similarities in the hydrogen bonds and C-H bonds absorption bands from the diagnostic region, between all the initial membranes, can be observed, including the control membrane without pollen and selenite. Also similar for all initial membranes are the amide bands, characteristic for amino acids and protein fragments resulted by Maillard reactions, generically called melanoidins [14]. A small difference in this region is shown by BCKPol5 in Appendix A, with the first amide band at 1728 cm^−1^ being more intense than in the other membranes, suggesting more C=O and COOH groups. By purification, the amide bands disappear, and only a small band for structural hydroxyls is present in nanocelluloses around 1647 cm^−1^. The hydrogen bonds in the diagnostic region corresponds only to HO..H bonds in cellulose for the purified membranes, with a peak around 3343 cm^−1^. In the initial membranes, two different peaks are visible for BCKPol25, respectively, around 3346 cm^−1^ for HO..H and around 3289 cm^−1^ for -HN..H (Figure 21). By translating this hypothesis to SeNPs, the peak around 3262 cm^−1^ suggests more amino groups in SeNPs than hydroxyls. This fact is confirmed in the amide bands which have significant and higher intensities than the COOH band at 1722 cm^−1^. The purification of membranes liberates the cellulose structure and allows the vibrations of C-C, C-H, C-O, and C-O-H in the structural carbohydrates (STCHO) and polysaccharides regions with many sharp bands. The control nanocellulose BNC20pK shows fewer sharp bands than BNC20pKPol25 in Figure 21. The region around 800 cm^−1^ is specific to Se-O and Se=O bonds, as marked in Appendix A and evidenced in Figure 21 by comparison with the control and Pol25 membranes. SeNPs present in this region’s bands at 932, 891, 874, 817, and 777 cm^−1^, as well as in the initial membrane of Pol25 bands at 916, 866–870, 818, and 777 cm^−1^, are visible. The common bands can be assigned to Se-O vibration, while the particular bands of SeNPs at 932 cm^−1^ might be assigned to Se-C and Se-O-C, respectively, at 891 cm^−1^ by analogy with the C-O-C glycosidic bond in cellulose around 899 cm^−1^.

## 4. Discussion

As previously mentioned, to the best of our knowledge, this is the first study of SeNPs biosynthesis with a Kombucha consortium. In selecting the sodium selenite variable for the Response Surface Methodology, we took into account different previously reported concentrations of sodium selenite that led to biosynthesis of SeNPs with different bacterial or fungal species. SeNPs have been produced by *Lactobacillus casei* using 200 µg/mL sodium selenite [42]. In another study with *Lactobacillus paracasei*, a concentration of 4 mM sodium selenite was selected [64]. *Lactobacillus acidophilus* was used for the SeNPs biosynthesis with 15 mM sodium selenite [40]. Another study with *Enterococcus faecalis* mentioned the range 0.19–2.97 mM sodium selenite for the production of biogenic SeNPs [65]. Other authors produced SeNPs using *Proteus mirabilis* with sodium selenite in the range of 1–5 mM [66]. There is another study with *Saccharomyces cerevisiae* which led to the production of SeNPs with 5–25 µg sodium selenite per 200 mL, and uniform SeNPs were observed at the higher tested concentration [43]. As the concentrations mentioned above are very diverse, we chose a wide range of sodium selenite in order to be certain that an effect can be observed. Regarding pollen concentration, 50 g/L pollen resulted in improved Kombucha fermentation, as previously reported [5]; therefore, we ranged around this value. 

The formation of SeNPs could be visually observed by the change in tea color and turbidity. The formed SeNPs have a Se-based core with an average diameter of approximately 50 nm based on TEM-EDX. 

The DLS data indicate that few aggregates/large particles are present, but their proportion is negligible. The transformation from intensity to volume and number assumes that the NP are spherical which seems to be the case based on TEM. The “biocorona” could induce some deviations from this sphericity, as well as in the refractive index; therefore, the transformation needs to be interpreted with care. The closest to reality is probably the volume distribution. The mean diameter around 50 nm, is in accordance with TEM, and suggests a thin layer formed by the ”biocorona” in these SeNPs, such as one formed by polyphenols, short oligosachharides, peptides, and other relatively small compounds. The bigger SeNPs probably have a ”biocorona” that includes proteins and larger oligosaccharides, which could explain the difference between TEM and DLS. Based on the diameters and volumes occupied, the population of the larger SeNPs seem to be approximately 75% of the population of smaller SeNPs. The number distribution seems to be biased to the smaller SeNPs.

Brick-red elemental selenium Se^0^ is unstable, and SeNPs quickly aggregate into gray or black particles in the absence of a stabilizing agent. The stabilization of SeNPs was previously shown to drastically depend on the hydrogen bonding environment around Se^0^, the disruption of Se..HO interactions with NaOH or dimethylsulfoxide, leading to precipitation [67]. Therefore, besides the various reducing agents used to obtain Se^0^, a stabilizing agent or molecular system with the propensity towards hydrogen bonds formation like -OH, -COOH, and -NH_2_ groups is necessary. This hypothesis was tested in the mentioned study [67] by comparing the nonstabilized SeNPs obtained by Na_2_SeO_3_ reduction with 0.2 M ascorbic acid solution with the same SeNPs further stabilized with lentinan, a one-branched β-(1,3)-D-glucan extracted from the fruiting bodies of *Lentinus edodes*. The nonstabilized SeNPs aggregated and precipitated, while the lentinan-SeNPs remained stable as an orange–red solution. The presence of the hydrogen bonds in lentinan-SeNPs was proved by FTIR spectroscopy in the diagnostic region with a strong band centered on 3388 cm^−1^, while the importance of H-bonding and possible van de Waals interactions in stabilizing the SeNPs was proved by using solvents with strong polarity like NaOH and dimethylsulfoxide [67], as previously mentioned.

Indeed, FTIR spectroscopy is usually employed to evaluate the hydrogen bonding and to investigate the molecular fingerprints at particular wavenumbers. Our SeNPs showed a wide absorption band centered around 3262 cm^−1^ that can be seen as a convolution of the hydrogen bonds vibrations corresponding to -OH, -COOH, and -NH_2_ groups. The hydrogen bonds might represent the first energetic layer between Se^0^ and ”biocorona” that assures stabilization against aggregation. Figure 7b, showing the comparative spectra of SeNPs, gallic acid, catechin and BSA, as well as Appendix A with more compounds, suggest that the ”biocorona” is similar to both BSA and gallic acid, with the main peak centered at 3282 cm^−1^, and catechin, with the centered band at 3265 cm^−1^ for the polyphenol structure [68]. The amide I, II, and III bands at 1638, 1541, and 1450 cm^−1^ presented in Figure 6 and Figure 7 are similar to the bands at 1656, 1548, and 1450 cm^−1^ obtained by actinobacterial reducing of Na_2_SeO_3_ and assigned to the secondary structure of proteins [69]. Moreover, the band at 1082 cm^−1^ in our SeNPs assigned to C-N bond vibration in amino acids, amines, and amides appeared around 1077 cm^−1^ in biosynthesized SeNPs using H_2_SeO_3_ and leaves extracted from *Withania somnifera* [70]. At 1024 cm^−1^, we assigned the C-O [71] to phenolic nature of SeNPs ”biocorona”, appearing also in catechin at 1026 cm^−1^. The absorption bands observed at 1541, 932, 891, 874 cm^−1^, and 777 cm^−1^ in Figure 7 and assigned to possible Se-O, Se-C, and Se-N bonds in the proteic and hydroxycinnamic ”biocorona” might correlate with the new band around 773 cm^−1^ in SeNPs obtained with lemon plant extract [71] and with the ones at 814, 775, 710, 677, and 613 cm^−1^ in SeNPs biosynthesized from Na_2_SeO_3_ with *Terminalia arjuna* leaf extract [72]. Moreover, the vibration of the Se-C bond was assigned around 750 cm^−1^ in a S-Se-C quantum dots system [73], while the Se=C vibration was placed between 800–775 cm^−1^ in selenosemicarbazones [74], which might help the understanding of various bonds and molecular structures stabilizing the biogenic SeNPs. As the polyphenols underwent oxidation, it is possible that the FTIR bands were influenced as well. The contact with SeNPs could also influence the FTIR bands.

The XRD pattern of SeNPs obtained by Na_2_SeO_3_ bioreduction with an actinobacterial strain evidenced a number of peaks around 14°, 25°, 31°, 42°, 44°, 44°, 52°, and 56° but without particular assignments [69]. Instead, amorphous SeNPs with the main peak centered on 25° were obtained by green synthesis using H_2_SeO_3_ and *Withania somnifera* leaf extract [70]. The diffractogram of KPol25 contains both amorphous region and additional peaks at 13.68°, 17.26°, 17.88°, 18.74°, 26.98°, 33.10°, 33.96°, and 43.94° that might suggest SeNPs and organo–selenium compounds. SeNPs obtained by the reduction of Na_2_SeO_3_ with *Terminalia arjuna* leaf macerate evidenced diffraction peaks at 23.59°, 29.78°, 41.29°, 43.67°, 45.84°, 51.74°, 56.13°, and 61.81° [72].

The content of soluble polyphenols, flavonoids, and hydroxycinnamic acids decreased in the Kombucha beverages following the increase in selenite concentration and the formation of SeNPs (Figure 1 and Figure 5). The most probable explanation is that they are involved in sodium selenite reduction and SeNPs stabilization. In the case of the KPol25 sample, the concentrations of phenolic compounds tend to increase compared with KPol15, and for the flavonoid content, the difference with respect to K was only marginally significant. This indicates that the polyphenols are released from pollen by fermentation, and that at 25 g pollen, the polyphenols content becomes in excess with respect to the amount necessary for complete selenite reduction and SeNPs formation. The necessity and involvement of polyphenols from pollen in the formation of SeNPs is deduced from the *p*-value < 0.0001 in Table 3. The SeNPs cannot be efficiently formed in simple black tea Kombucha beverage. Although in most cases a higher content of phenolic compounds is associated with a higher antioxidant activity, at 190 mg selenit, it can be seen that the AOA is less affected than the content of polyphenols. This is due to the formation of SeNPs, which have significant AOA. as can be seen in Figure 9. In the case of the KPol25 sample where the biosynthesis yield of SeNPs was over 95%, there are no significant AOA differences compared with K. The AOA most affected is at 15 g pollen, probably because there is still a significant amount of selenite, a lower proportion of SeNPs, and a minimum of soluble polyphenols available in the reduced form. 

An additional effect of increasing pollen concentration was the increase in the lactic acid content. This is probably the result of the stimulation of lactic acid bacteria from the initial Kombucha consortium and of the contribution with fructophilic lactic acid bacteria (FLAB) from pollen [75,76]. The addition of pollen contributes with both FLAB and polyphenols, the latter being able to function as electron acceptors [77]. The obligatory FLAB grows poorly on glucose in the absence of electron acceptors. The facultative FLAB grows slower on glucose in the absence of electron acceptors [75]. At 5 g pollen, the content of polyphenols released is probably not enough to allow FLAB to grow on glucose, and most of them consume fructose. This explains the drop in the D-fructose level in KPol5 compared with K. The increase in glucose content in KPol5 compared with KPol indicates that a new microbial equilibrium is established, dominated by fructose-consuming species. In the case of KPol15 and KPol25, with 3× and 5×, respectively, more pollen added than in KPol5, we propose that the increase in the release of polyphenols allows FLAB to consume glucose as well, using polyphenols as electron acceptors and/or changes the population dynamics, this time towards glucose-consuming species. This can explain the tendency of glucose decrease and fructose increase in KPol15 and KPol25. The decrease in TPC, especially HAT, is therefore a consequence of both selenite reduction with SeNPs formation and polyphenols involvement in microbial metabolism. 

Using LC-MS, we identified a series of compounds in black tea, Kombucha beverages, and pollen extracts, including polyphenols fatty acids, two amino acids, γ-aminobutyric acid (GABA) and valine, and one polyamine (spermidine) (Table 4). The smallest compound, GABA, was identified in all the Kombucha samples with pollen and in pollen extracts with methanol or ethanol. GABA is produced by plants, microorganisms, and animals and is implicated in numerous biological activities. GABA has been recently shown to be produced by a strain of *Levilactobacillus brevis* isolated from green tea leaves [78]. As we did not detect GABA in Kombucha without pollen, its presence suggests as origin the pollen. GABA homeostasis was previously shown to regulate pollen germinationin *Picea wilsonii* and growth of pollen tubes in various plants [79,80]. Pollen has been used as a natural catalyst to produce GABA from monosodium glutamate [81]. Valine was found in the same samples as GABA. Valine is a common amino acid found in pollen, along other essential amino acids [82]. Spermidine appeared in Kombucha with pollen but not in pollen extracts. It probably resulted from the microbial decarboxylation of amino acids [83]. As pollen came with significant content of amino acids, the level of polyamines increased such that it became detectable by LC-MS. Caffeine is a known alkaloid found in significant amounts in black tea, and it appeared in all Kombucha variants as well. 

Sinapic acid was detected only in KPol. It is possible that it could be extracted only enzymatically from pollen and that at higher selenite concentrations it is transformed. It was previously reported in pollen [84] and Assam tea [85].

In the case of (dihydro)-resveratrol, which seems to be present only in black tea, the LC-MS data gave an average *m/z* (230.2) between ionized resveratrol (229.2) and ionized dihydro-resveratrol (231.2). Resveratrol has been previously reported in green tea [86] and in black tea with values between 51–60 mg/kg [87]. As this is a compound with known high biological activity, we used transresveratrol (t-resveratrol) reference material to detect and quantify it by HPLC-DAD (Appendix A). The retention time of the reference material (7.015 min) was close to a retention peak from the sample (6.980 min) and, based on the calibration curve, the quantity determined was 32.8 ± 2.0 mg/kg. In LC-MS, the *m/z* for t-resveratrol reference material was 229.2, as expected. Dihydro-resveratrol (DHR) was proposed to be a gut-microbiota-derived metabolite formed by the metabolization of resveratrol [88]. We shall continue investigations to see if DHR is present as well in black tea and in which quantities. 

The fatty acids (FA) palmitic acid and linolenic acid were detected only in pollen extracts in our study, being already previously reported in pollen [89,90]. Under Kombucha fermentation, they are probably too hydrophobic to be extracted in tea or completely metabolized. It was previously stated that the pollens high in oleic and palmitic acids are especially important in honey bee nutrition, and linolenic acid would be important for growth inhibition of the spore-forming bacteria such as *Paenibacillus larvae* (American foulbrood) and *Melissococcus pluton* (European foulbrood) [89]. The FA composition of pollen was previously shown to depend on the harvest season [90].

Epigallocatechin gallate (EGCG) is mainly found in green tea and yellow tea, but it was reported in black, oolong, and white tea as well and is one of the major polyphenols in tea [91,92]. This compound is metabolized during Kombucha fermentation into epigallocatechin (EGC) by mechanisms yet unclear [93]. In our case, EGCG is detected in black tea and disappears in all fermented samples, but EGC is detected only in the presence of the highest content of pollen. The absence of EGC, in all other fermentation samples could be due to its further metabolization, degradation, and/or to its oxidation as a result of implication in selenite reduction. EGC could be sourced from pollen, and/or it is possible for there to be too much pollen to inhibit the growth of specific microorganisms or to modulate the activity of enzymes and/or other compounds implicated in EGC metabolization. There are only a few studies reporting the presence of EGC in pollen, mainly in tea pollen, the compound being mainly released by a combination of enzymatic and ultrasonic treatments [94].

Rutin (quercetin-3-O-rutinoside) is a flavonoid known for its high various biological activities [95]. Rutin was identified by LC-MS only in black tea and, due to its importance, we confirmed and quantified it using HPLC-DAD, the values being somewhat lower (2.4 ± 0.1 mg/L) than previously reported: 0.60–3.35 mg/100 mL [96] and 3.3–6 mg/L [97] in black teas and tea products. The compound was not detected in Kombucha beverages, probably because of microbial metabolization, degradation, and/or oxidation by selenite.

The soluble Si content presents a particular behavior. High selenite content induces the need for high SCOBY inoculum to increase the Si release from pollen. This is understandable due to selenite toxicity. The system presents an interesting behavior at low selenite concentration as well as at high SCOBY concentration, respectively. At low selenite concentration, the lower the SCOBY inoculation, the higher the soluble Si. At high SCOBY concentration, the higher the selenite concentration, the higher the soluble Si content. Our hypothesis is that increased SCOBY inoculum on one hand releases significant amounts of soluble Si by pollen fermentation and, on the other hand, induces an acidic pH, which reduces Si solubility from the 1–2 mM at neutral pH [98,99]. The result is an accelerated polycondensation and precipitation of Si at higher SCOBY inoculum than at lower SCOBY inoculum. It is also possible that SCOBY contributes to lower soluble Si by actively metabolizing the soluble Si into insoluble forms, but this needs experimental confirmation.

Several studies claimed that Se has a narrow physiological window, the difference between the beneficial and toxic dose being less than an order of magnitude [34]. In our study, the SeNPs led to an increased cell metabolism at low concentrations, 0.5–2.5 µg/mL, and at the highest concentration tested, 10 µg/mL, SeNPs became slightly cytotoxic (Figure 11). By measuring ROS production, our results indicate the capacity of SeNPs to re-establish ROS homeostasis (Figure 13). The Kombucha beverages KPol15 and KPol25 increased cell metabolism significantly (Figure 14). A slight pro-oxidant effect compared with K was observed for KPol5 and KPol15 samples at 3 and 5 mg/mL (Figure 15), where sodium selenite was not completely reduced. In sample KPol15, where sodium selenite was converted approximately 50% into SeNPs, ROS scavenging slightly increased by increasing the sample concentration to 7 mg/mL. The KPol25 sample, which presented the highest SeNPs biosynthesis yield, induced the highest ROS scavenging capacity of all analyzed samples. Combined with the increased cell metabolism, our data indicate that the proposed product is suitable for biomedical applications.

Previous studies have only investigated SeNPs on dermal fibroblasts and other cell types but not on gingival fibroblasts, to the best of our knowledge. Moreover, most studies reported until now investigated mainly the cytocompatibility at high concentrations, useful for antimicrobial applications [100,101,102,103], with limited information at low concentrations, which seem to have similar effects as SeNPs developed in this research but on dermal fibroblasts [104]. Our study is the first to report SeNPs and SCOBY beverage with SeNPs that induce a significant metabolic increase and antioxidant activity in gingival fibroblasts at low concentrations. We propose that our SeNPs-based products can be applied at low doses as formulations with antimicrobial products for a complementary or synergic effect.

Besides SeNPs and Kombucha beverage enriched in biogenic SeNPs, fermented pollen is another obtained product that could be considered a functional food with increased bioavailability and fortified in Se under the form of SeNPs. The system also stimulates the formation of bacterial cellulose membrane, which indicates the stimulation of acetic bacteria activity as well, besides the stimulation of lactic acid bacteria. 

The whiter appearance of bacterial cellulose membrane BCKPol25 compared with BCK is probably the result of lower availability of polyphenols and proteins to be involved in melanoidin formation, as they are involved in SeNPs formation and stabilization. The higher crystallinity of formed nanocellulose with a high level of pollen suggests the suitability of this type of nanocellulose for applications where increased mechanical strength is needed. 

## 5. Conclusions

In this study, SeNPs were obtained by Kombucha fermentation with pollen, resulting as well in a new Kombucha–pollen beverage enriched in selenium, with antioxidant and normal cell proliferation properties. TEM-EDX analysis revealed quasi-spherical SeNPs of 50 nm average, confirmed by DLS, which revealed also a second population with 138 nm average diameter. FTIR analysis suggested the presence of polyphenol and protein ”biocorona”, stabilized by hydrogen bonds. Both selenite and pollen positively modulate the microbial population. Higher and whiter bacterial cellulose membrane and fermented pollen enriched in SeNPs were obtained. XRD analysis evidenced a 1β-cellulose reinforced structure in the case of bacterial cellulose enriched with the highest SeNPs content. Cell viability and proliferation assays indicated a high degree of biocompatibility with potential of stimulating proliferation of the human gingival fibroblast cell line (HGF-1). Furthermore, the in vitro antioxidant activity assays revealed the potential of SeNPs and Kombucha beverage enriched with in situ biosynthesized SeNPs to re-establish ROS homeostasis.

## Figures and Tables

**Figure 1 antioxidants-12-01711-f001:**
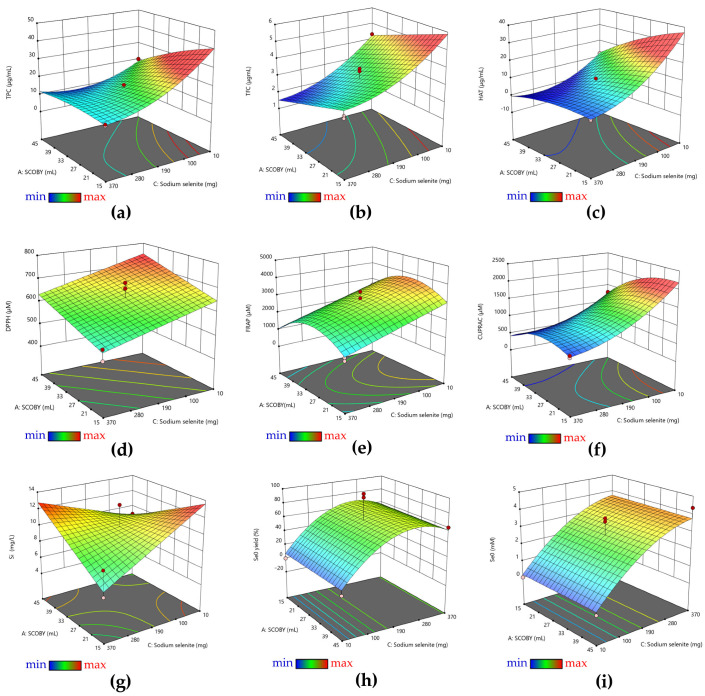
Three-dimensional representations of the influence of the three variables on the properties of pollen-fermented Kombucha beverage, as well as on maximizing the production of biogenic SeNPs: (**a**) TPC; (**b**) TFC; (**c**) HAT; (**d**) DPPH; (**e**) FRAP; (**f**) CUPRAC; (**g**) Silicon content; (**h**) Se^0^ biosynthesis yield; and (**i**) Se^0^ content. The color gradient corresponds to the range of the response variables (min to max), which are presented in Appendix A. The dots highlight the experimental points.

**Figure 2 antioxidants-12-01711-f002:**
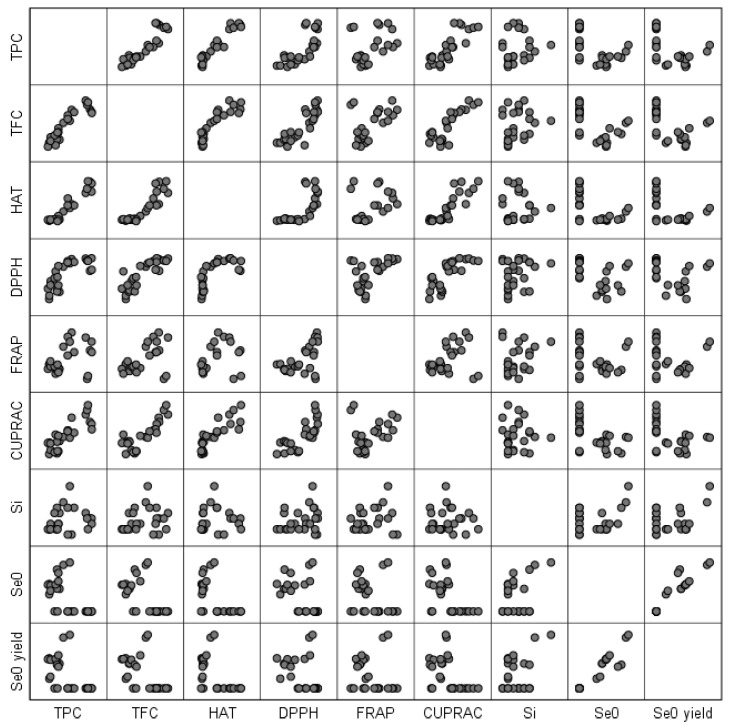
Pearson correlation plot for the responses of the RSM experimental design.

**Figure 3 antioxidants-12-01711-f003:**
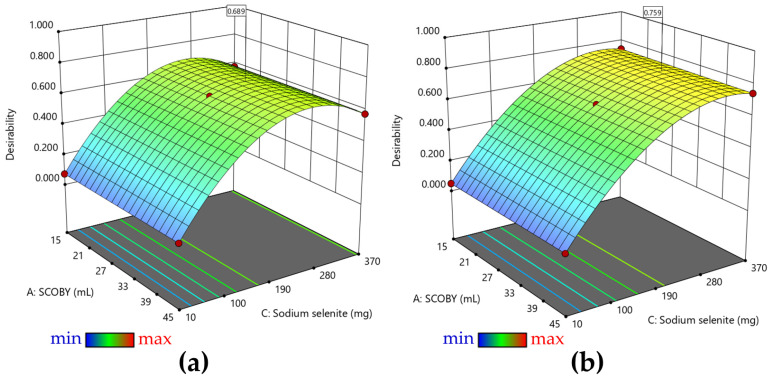
Desirability: (**a**) 3D representation of the desirability function for maximizing Se^0^ yield and (**b**) 3D representation of the desirability function for maximizing Se^0^ yield and Se^0^ content. The color gradient corresponds to the range of the desirability function (min = 0 to max = 1). The dots highlight the predicted points.

**Figure 4 antioxidants-12-01711-f004:**
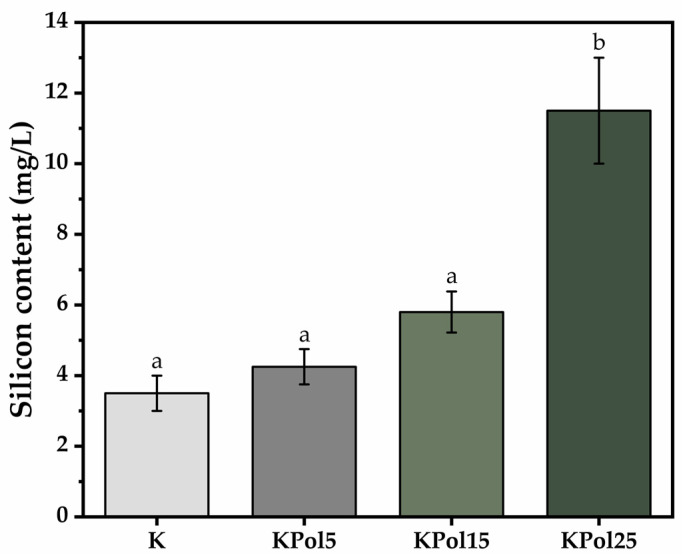
Soluble silicon content; K—Kombucha beverage with 30 mL SCOBY; KPol5—Kombucha beverage with 30 mL SCOBY, 5 g pollen, and 190 mg sodium selenite; KPol15—Kombucha beverage with 30 mL SCOBY, 15 g pollen, and 190 mg sodium selenite; KPol25—Kombucha beverage with 30 mL SCOBY, 25 g pollen, and 190 mg sodium selenite (±error bars, α < 0.05, n = 3 for K, n = 2 for Kpol5, KPol25, and n = 5 for KPol15, different letters indicate statistically significant differences between samples).

**Figure 5 antioxidants-12-01711-f005:**
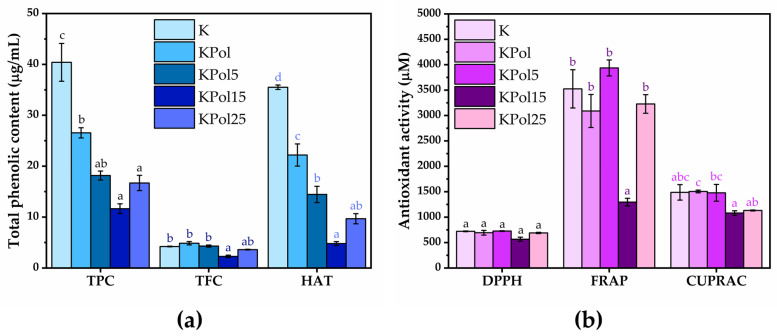
Influence of Kombucha modulation on total phenolic content and antioxidant activity: (**a**) total phenolic content; (**b**) antioxidant activity. K—Kombucha beverage with 30 mL SCOBY; KPol—Kombucha beverage with 30 mL SCOBY, 15 g pollen, and 10 mg sodium selenite; KPol5—Kombucha beverage with 30 mL SCOBY, 5 g pollen, and 190 mg sodium selenite; KPol15—Kombucha beverage with 30 mL SCOBY, 15 g pollen, and 190 mg sodium selenite, KPol25—Kombucha beverage with 30 mL SCOBY, 25 g pollen, and 190 mg sodium selenite (±error bars, α < 0.05, n = 3 for K, n = 2 for KPol, Kpol5, KPol25, and n = 5 for KPol15, different letters indicate statistically significant differences between samples).

**Figure 6 antioxidants-12-01711-f006:**
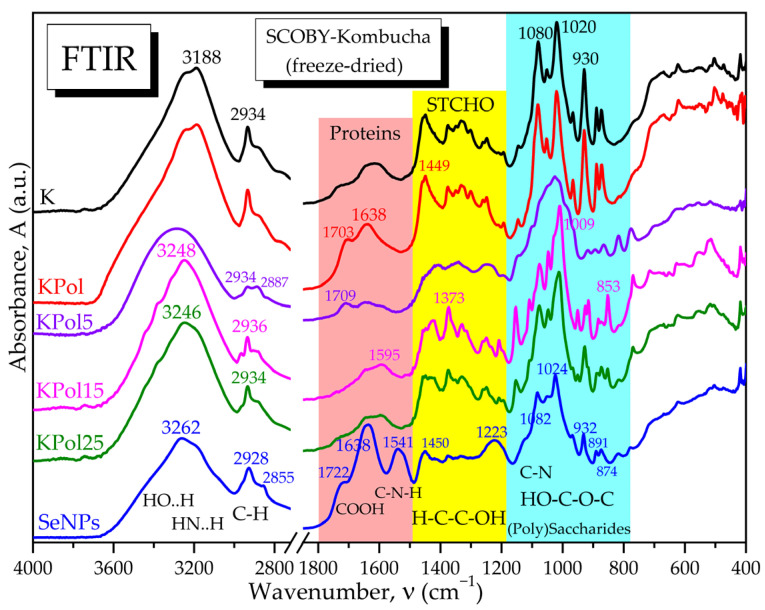
ATR-FTIR spectra of freeze-dried samples of SCOBY-Kombucha beverage and SeNPs: K—Kombucha beverage with 30 mL SCOBY; KPol—Kombucha beverage with 30 mL SCOBY, 15 g pollen, and 10 mg sodium selenite; KPol5—Kombucha beverage with 30 mL SCOBY, 5 g pollen, and 190 mg sodium selenite; KPol15—Kombucha beverage with 30 mL SCOBY, 15 g pollen, and 190 mg sodium selenite; KPol25—Kombucha beverage with 30 mL SCOBY, 25 g pollen, and 190 mg sodium selenite; SeNPs—SeNPs Kombucha.

**Figure 7 antioxidants-12-01711-f007:**
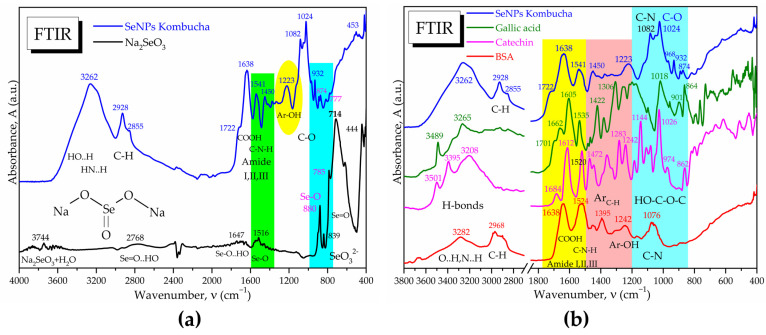
ATR-FTIR spectrum of SeNPs compared with (**a**) Na_2_SeO_3_; (**b**) comparison of SeNPs spectra with those of gallic acid, catechin hydrate, and BSA.

**Figure 8 antioxidants-12-01711-f008:**
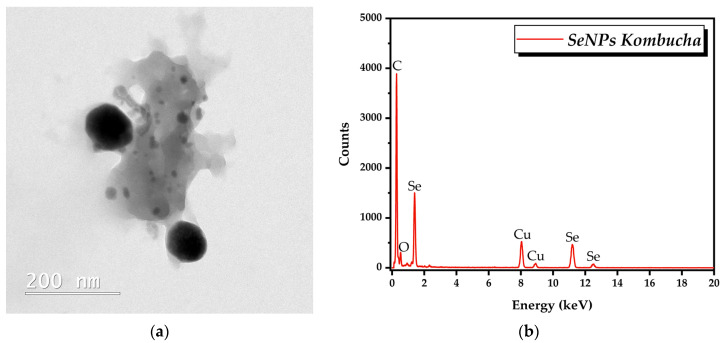
TEM-EDX analysis of SeNPs: (**a**) TEM analysis; (**b**) EDX analysis.

**Figure 9 antioxidants-12-01711-f009:**
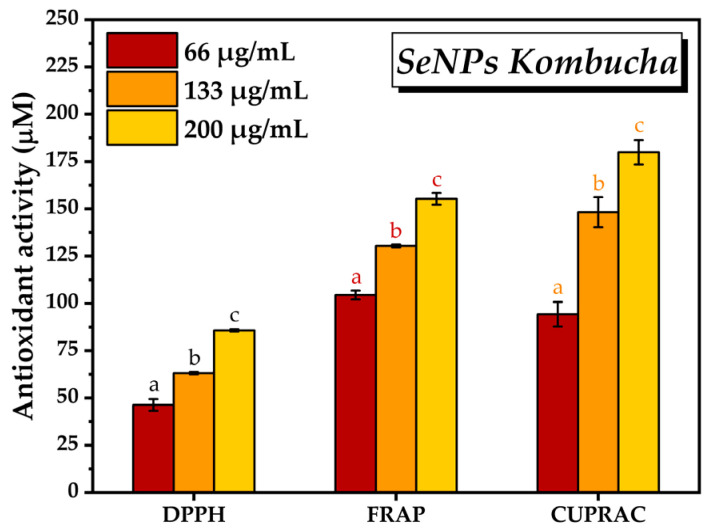
Antioxidant activity of SeNPs by DPPH, FRAP, and CUPRAC assays (±error bars, α < 0.05, and n = 3, different letters indicate statistically significant differences between samples). The antioxidant activity was expressed as µM Trolox Equivalents (TE).

**Figure 10 antioxidants-12-01711-f010:**
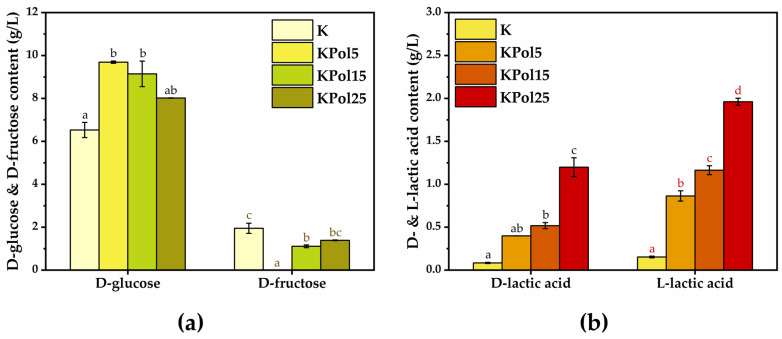
Dynamics of bacteria and yeast populations: (**a**) D-glucose and D-fructose content; (**b**) D-lactic acid and L-lactic acid content. K—Kombucha beverage with 30 mL SCOBY; KPol5—Kombucha beverage with 30 mL SCOBY, 5 g pollen, and 190 mg sodium selenite; KPol15—Kombucha beverage with 30 mL SCOBY, 15 g pollen, and 190 mg sodium selenite; KPol25—Kombucha beverage with 30 mL SCOBY, 25 g pollen, and 190 mg sodium selenite (±error bars, α < 0.05, n = 3 for K, n = 2 for Kpol5, KPol25, and n = 5 for KPol15, different letters indicate statistically significant differences between samples).

**Figure 11 antioxidants-12-01711-f011:**
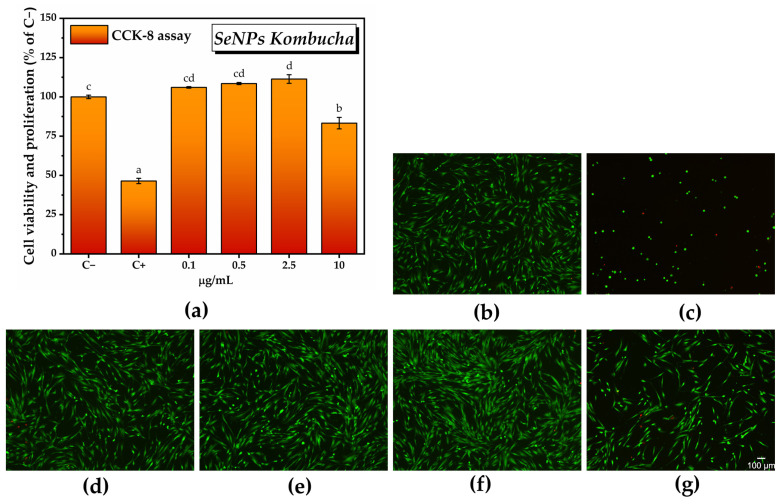
Biocompatibility of SeNPs: (**a**) CCK-8 assay (± error bars, α < 0.05, and n = 3, different letters indicate statistically significant differences between samples); (**b**–**g**) LIVE/DEAD assay (live cells—green fluorescence, dead cells—red fluorescence): (**b**) untreated cells; (C−; negative cytotoxicity control); (**c**) cells treated with 7.5% DMSO (C+; positive cytotoxicity control); (**d**) cells treated with 0.1 µg/mL SeNPs Kombucha; (**e**) cells treated with 0.5 µg/mL SeNPs Kombucha; (**f**) cells treated with 2.5 µg/mL SeNPs Kombucha; and (**g**) cells treated with 10 µg/mL SeNPs Kombucha.

**Figure 12 antioxidants-12-01711-f012:**
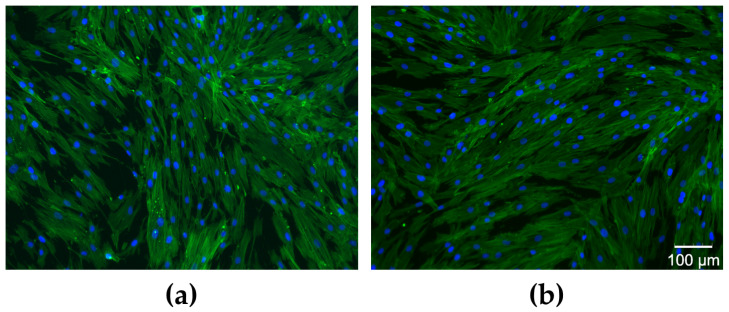
Cell morphology–Fluorescence microscopy images after actin cytoskeleton labeling with Alexa Fluor 488-coupled phalloidin (green fluorescence) and the nuclei with DAPI (blue fluorescence): (**a**) untreated cells and (**b**) cells treated with 2.5 µg/mL SeNPs.

**Figure 13 antioxidants-12-01711-f013:**
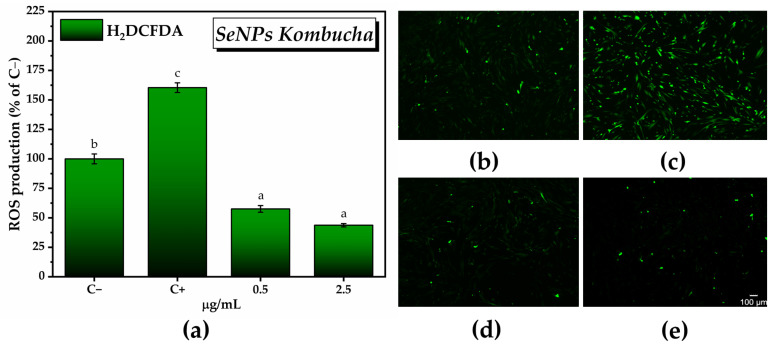
In vitro antioxidant activity of SeNPs: (**a**) Labeling and quantifying total ROS with H_2_DCFDA(±error bars, α < 0.05, and n = 3, different letters indicate statistically significant differences between samples); (**b**–**e**) Fluorescence microscopy images after labeling total ROS with H_2_DCFDA (green fluorescence): (**b**) untreated cells (C−; negative control); (**c**) cells treated with 37 µM H_2_O_2_ (C+; positive control; ROS inducer); (**d**) cells treated with 0.5 µg/mL SeNPs Kombucha in the presence of ROS inducer; and (**e**) cells treated with 2.5 µg/mL SeNPs Kombucha in the presence of ROS inducer.

**Figure 14 antioxidants-12-01711-f014:**
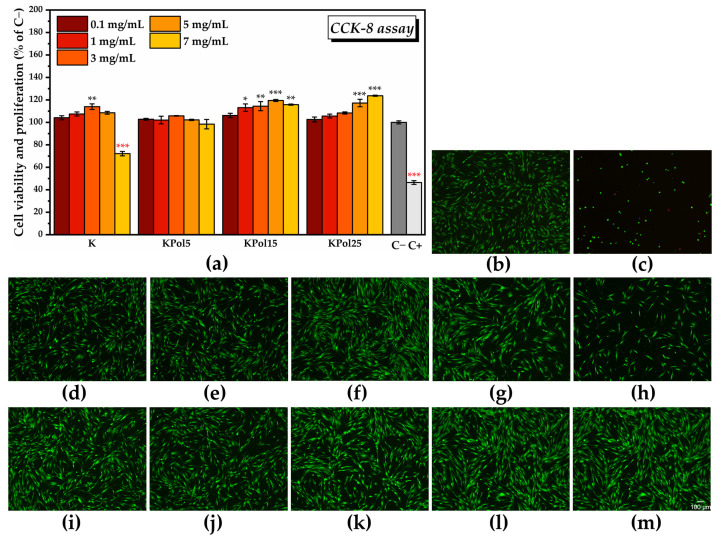
Biocompatibility of Kombucha beverage: (**a**) CCK-8 assay (±error bars, α < 0.05, n = 3, *—σ between 0.05 and 0.01, **—σ between 0.01 and 0.001, and ***—σ < 0.001; black stars indicate statistically significant values that exceed C−; and red stars indicate statistically significant values that are below C−); (**b–m**) LIVE/DEAD assay (live cells—green fluorescence, dead cells—red fluorescence): (**b**) untreated cells (C−; negative cytotoxicity control); (**c**) cells treated with 7.5% DMSO (C+; positive cytotoxicity control); HGF-1 cells treated with: (**d**) 0.1 mg/mL K; (**e**) 1 mg/mL K; (**f**) 3 mg/mL K; (**g**) 5 mg/mL K; (**h**) 7 mg/mL K; (**i**) 0.1 mg/mL KPol25; (**j**) 1 mg/mL KPol25; (**k**) 3 mg/mL KPol25; (**l**) 5 mg/mL KPol25; (**m**) 7 mg/mL KPol25. K—Kombucha beverage with 30 mL SCOBY; KPol5—Kombucha beverage with 30 mL SCOBY, 5 g pollen, and 190 mg sodium selenite; KPol15—Kombucha beverage with 30 mL SCOBY,15 g pollen, and 190 mg sodium selenite; and KPol25—Kombucha beverage with 30 mL SCOBY, 25 g pollen, and 190 mg sodium selenite.

**Figure 15 antioxidants-12-01711-f015:**
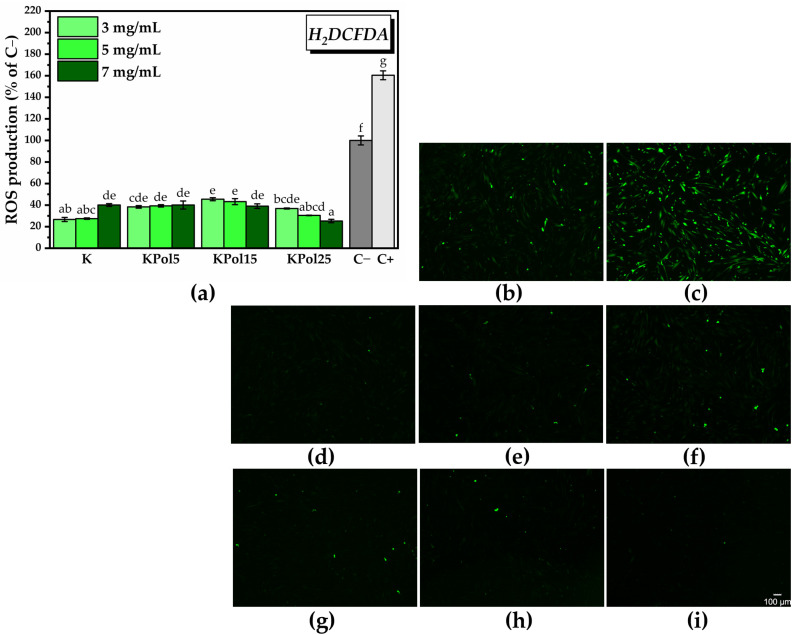
In vitro antioxidant activity of Kombucha beverage: (**a**) labeling and quantifying total ROS with H_2_DCFDA (±error bars, α < 0.05, and n = 3, different letters indicate statistically significant differences between samples); (**b–i**) fluorescence microscopy images after labeling total intracellular ROS with H_2_DCFDA (green fluorescence): (**b**) untreated cells (C−; negative control); (**c**) cells treated with 37 µM H_2_O_2_ (C+; positive control; ROS inducer); HGF-1 cells incubated in the presence of ROS inducer and (**d**) 3 mg/mL K; (**e**) 5 mg/mL K; (**f**) 7 mg/mL K; (**g**) 3 mg/mL KPol25; (**h**) 5 mg/mL KPol25; and (**i**) 7 mg/mL KPol25. K—Kombucha beverage with 30 mL SCOBY; KPol5—Kombucha beverage with 30 mL SCOBY, 5 g pollen, and 190 mg sodium selenite; KPol15—Kombucha beverage with 30 mL SCOBY, 15 g pollen, and 190 mg sodium selenite; KPol25—Kombucha beverage with 30 mL SCOBY, 25 g pollen, and 190 mg sodium selenite.

**Figure 16 antioxidants-12-01711-f016:**
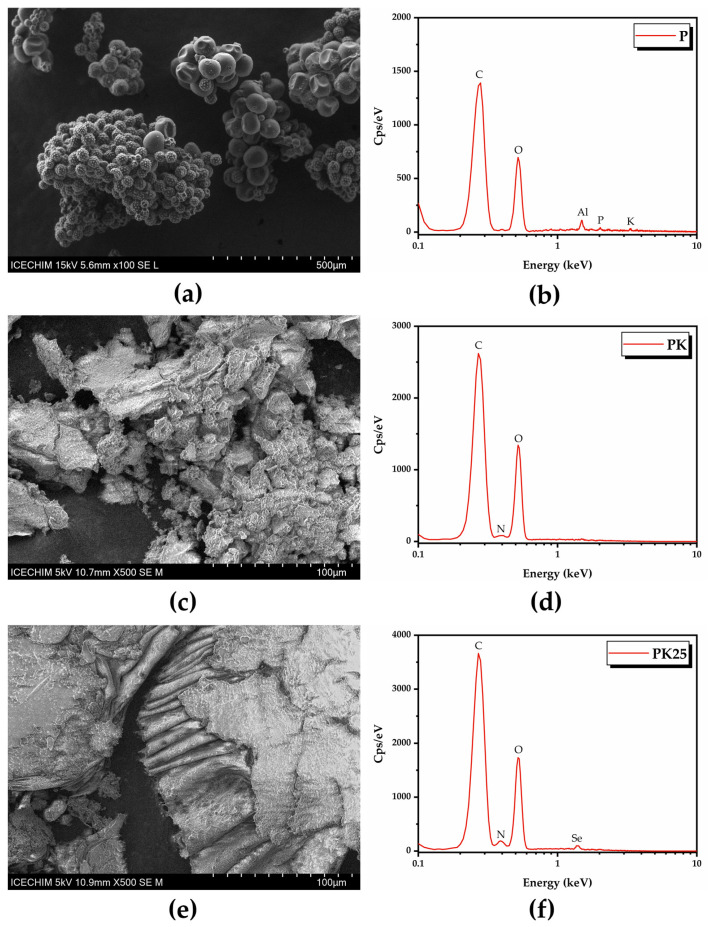
SEM-EDX analysis of fresh polyfloral pollen, and fermented pollen: (**a**) P; (**b**) EDX spectrum of P; (**c**) PK; (**d**) EDX spectrum of PK; (**e**) PK25; (**f**) EDX spectrum of PK25. P—fresh polyfloral pollen; PK—fermented pollen from KPol (30 mL SCOBY, 15 g pollen, and 10 mg sodium selenite); PK25—fermented pollen from KPol25 (30 mL SCOBY, 15 g pollen, and 190 mg sodium selenite).

**Figure 17 antioxidants-12-01711-f017:**
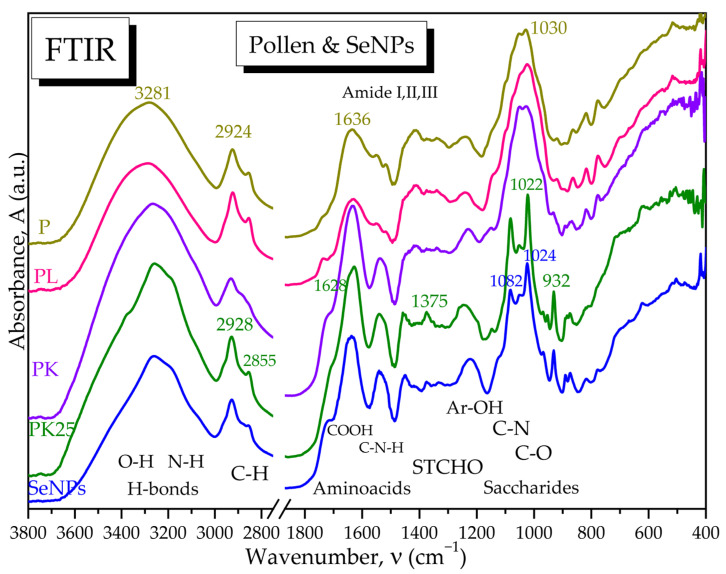
ATR-FTIR spectra of main pollen samples and SeNPs: P is raw pollen; PL is lyophilized pollen; PK is fermented pollen with SCOBY and 10 mg Na_2_SeO_3_; PK25 is fermented pollen with 190 mg Na_2_SeO_3_; and SeNPs are the selenium nanoparticles.

**Figure 18 antioxidants-12-01711-f018:**
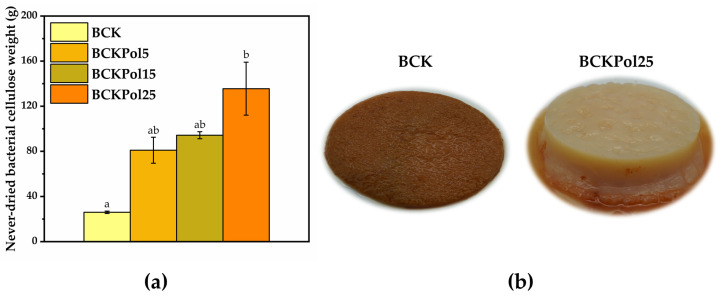
Analysis of never-dried bacterial cellulose in terms of weight and topography: (**a**) Never-dried bacterial cellulose weight (±error bars, α < 0.05, n = 3, different letters indicate statistically significant differences between samples). (**b**) Topographical analysis of BCK, and BCKPol25; BCK—bacterial cellulose from Kombucha beverage prepared with 30 mL SCOBY; BCKPol5—bacterial cellulose from Kombucha beverage prepared with 30 mL SCOBY, 5 g pollen, and 190 mg sodium selenite; BCKPol15—bacterial cellulose from Kombucha beverage prepared with 30 mL SCOBY, 15 g pollen, and 190 mg sodium selenite; and BCKPol25—bacterial cellulose from Kombucha beverage prepared with 30 mL SCOBY, 25 g pollen, and 190 mg sodium selenite.

**Figure 19 antioxidants-12-01711-f019:**
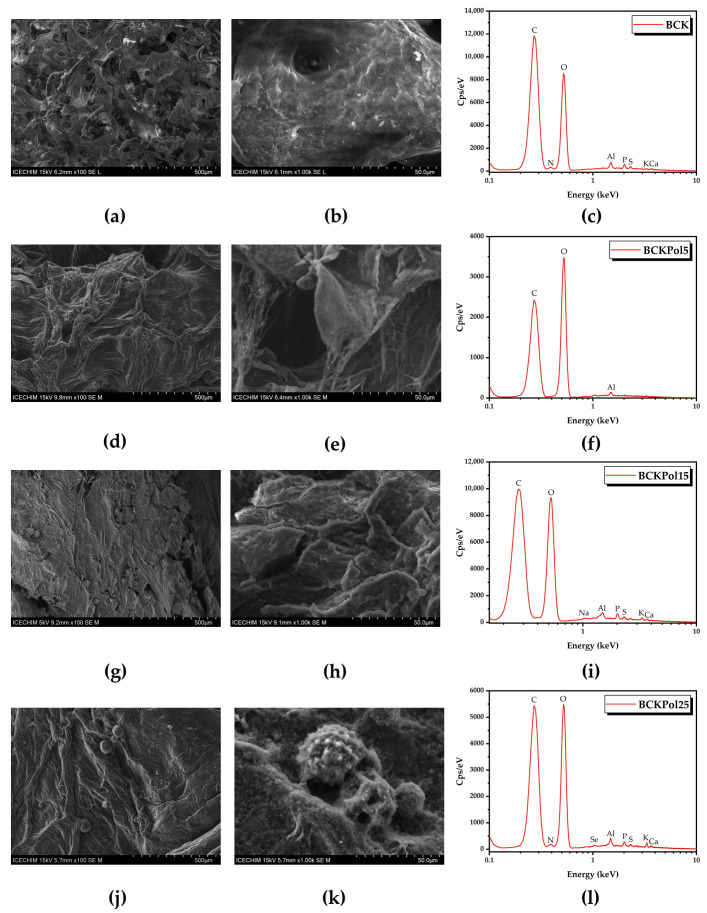
SEM-EDX analysis of BC: (**a**) BCK 100×; (**b**) BCK 1000×; (**c**) EDX spectrum of BCK; (**d**) BCKPol5 100×; (**e**) BCKPol5 1000×; (**f**) EDX spectrum of BCKPol5; (**g**) BCKPol15 100×; (**h**) BCKPol15 1000×; (**i**) EDX spectrum of BCKPol15; (**j**) BCKPol25 100×; (**k**) BCKPol25 1000×; (**l**) EDX spectrum of BCKPol25. BCK—bacterial cellulose from Kombucha beverage prepared with 30 mL SCOBY; BCKPol5—bacterial cellulose from Kombucha beverage prepared with 30 mL SCOBY, 5 g pollen, and 190 mg sodium selenite; BCKPol15—bacterial cellulose from Kombucha beverage prepared with 30 mL SCOBY, 15 g pollen, and 190 mg sodium selenite; and BCKPol25—bacterial cellulose from Kombucha beverage prepared with 30 mL SCOBY, 25 g pollen, and 190 mg sodium selenite.

**Figure 20 antioxidants-12-01711-f020:**
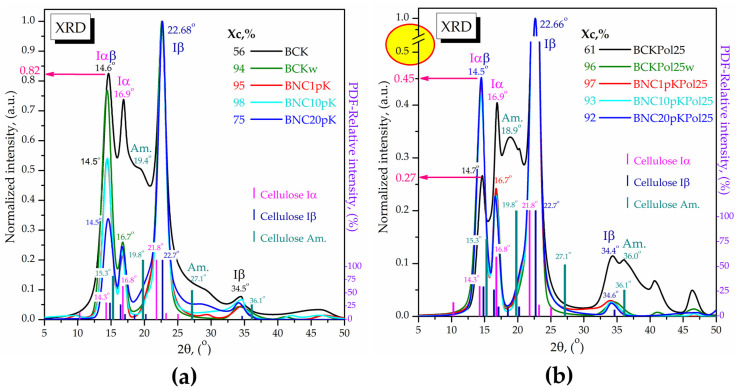
XRD analyses of bacterial celluloses membranes (BC) compared with the corresponding bacterial nanocelluloses (BNC) obtained by purification and microfluidization: (**a**) Control samples (without pollen and sodium selenite): BCK—bacterial cellulose from Kombucha beverage prepared with 30 mL SCOBY; BCKw—purified/washed bacterial cellulose; BNC1p—bacterial nanocellulose after 1 pass of microfluidization; BNC10pK—bacterial nanocellulose after 10 passes of microfluidization; BNC20pK—bacterial nanocellulose after 20 passes of microfluidization; (**b**) Samples from Kombucha modulation with pollen, and sodium selenite: BCPol25—bacterial cellulose from Kombucha beverage prepared with 30 mL SCOBY, 25 g pollen, and 190 mg of sodium selenite; BCKPol25w—purified/washed bacterial cellulose; BNC1pKPol25—bacterial nanocellulose after 1 pass of microfluidization; BNC10pKPol25—bacterial nanocellulose after 10 passes of microfluidization; and BNC20pKPol25—bacterial nanocellulose after 20 passes of microfluidization. The red circle filled with yellow in Figure 20b highlights the necessary scale break due to the high intensity of 22.66° peak.

**Figure 21 antioxidants-12-01711-f021:**
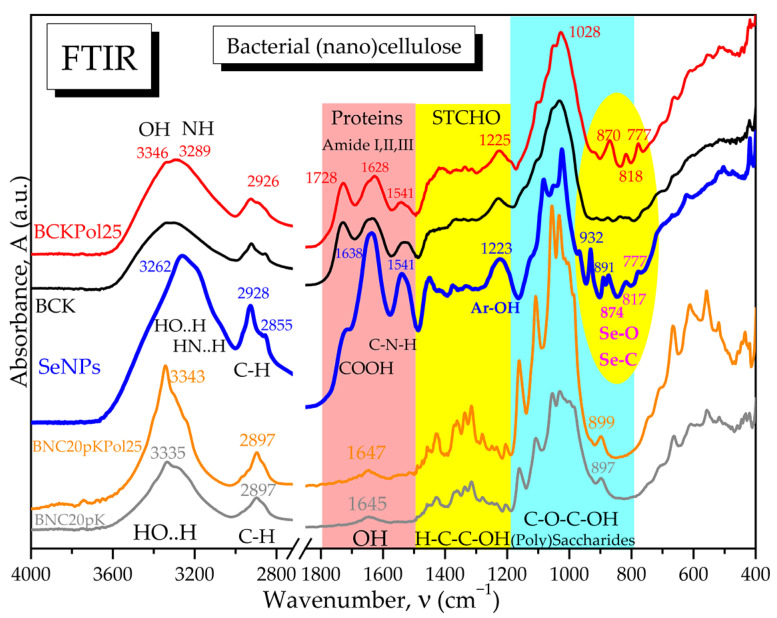
ATR-FTIR spectra: BCK—bacterial cellulose from Kombucha beverage prepared with 30 mL SCOBY (control); BNC20pK—bacterial nanocellulose (control) after 20 passes of microfluidization; BCKPol25—bacterial cellulose from Kombucha beverage prepared with 30 mL SCOBY, 25 g pollen, and 190 mg of sodium selenite; and BNC20pKPol25—bacterial nanocellulose after 20 passes of microfluidization.

**Table 1 antioxidants-12-01711-t001:** Variation interval of the three factors included in the experimental design.

Factor	Name	Units	Type	Minimum	Maximum	Coded Low	Coded High
A	SCOBY	mL	Numeric	15	45	−1	+1
B	Pollen	g	Numeric	5	25	−1	+1
C	Sodium selenite	mg	Numeric	10	370	−1	+1

**Table 2 antioxidants-12-01711-t002:** *p*-value from the ANOVA analysis (A-SCOBY, B-pollen, C-sodium selenite).

	Model	A	B	C	AB	AC	BC	A^2^	B^2^	C^2^
TPC	<0.0001	0.012	-	<0.0001	<0.0001	<0.0001	-	0.0261	<0.0001	<0.0001
TFC	<0.0001	0.0065	0.0009	<0.0001	0.0004	-	-	-	<0.0001	0.0057
HAT	<0.0001	0.9540	0.0288	<0.0001	<0.0001	0.0011	0.0859	0.0143	<0.0001	<0.0001
DPPH	<0.0001	0.0076	0.0415	0.0026	-	-	-	-	0.0003	-
FRAP	0.0005	0.3992	0.5170	0.0004	-	-	-	0.0034	0.0022	-
CUPRAC	<0.0001	0.9341	0.0033	<0.0001	0.0001	0.0243	0.0009	<0.0001	<0.0001	<0.0001
Si	0.0004	0.4258	<0.0001	0.0823	-	0.0068	-	-	-	-
Se^0^	<0.0001	-	<0.0001	<0.0001	-	-	<0.0001	-	-	0.0032
Se^0^ yield	<0.0001	-	<0.0001	0.0053	-	<0.0001	0.0083	-	-	<0.0001

**Table 3 antioxidants-12-01711-t003:** LC-TOF/MS analysis of phenolic content.

Compound Name	Black Tea	K	KPol	KPol5	KPol15	KPol25	P-MeOH	P-EtOH	P-NaOH	P-Acetone-Hexamethyl Tetramine-HCl
γ-Aminobutyric acid			104.1	104.1	104.1	104.1	104.1	104.1		
Valine			118.0	118.0	118.0	118.0	118.0	118.0		
Spermidine			146.1	146.1	146.1	146.1				146.0
Caffeine	195.0 *	195.0	195.1	195.0	195.0	195.0				
Sinapic acid			225.1							
(Dihydro)-resveratrol	230.2	230.2								
Palmitic acid						257.2	257.2	257.2		
Linolenic acid							279.2			
Epigallocatechin						305.1				
(−)-Epigallocatechin gallate	459.4									
Rutin	611.3									

* The numbers represent *m/z*.

**Table 4 antioxidants-12-01711-t004:** DLS analysis of SeNPs.

Mean Diameter (nm)	Intensity (%)	Volume (%)	Number (%)
49.06	-	-	100
51.39	-	34.34	-
135.68	-	65.66	-
431.06	100	-	-

## Data Availability

All the data are presented in this work.

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
