# Peer review of "Selenium-Fortified Kombucha–Pollen Beverage by In Situ Biosynthesized Selenium Nanoparticles with High Biocompatibility and Antioxidant Activity"

_antioxidants, 2023, doi:10.3390/antiox12091711_

Round 1
Reviewer 1 Report
It is an interesting paper with a lot of experimental data, but a reorganization is necessary because it is hard to follow its.
It is not clear how help the soluble silicon content to obtain a high content of selenium. Please clarify this. Which is the correlation between title of paper and the discussions about silicon (see also paragraph from lines 577 - 587).
In the sections 2.3.2 and 2.3.3, SeNPs not appears and is not clear how the experiments occurred related to these particles.
It is not clear why the caffeic acid / hydroxycinnamic acids were chosen for FTIR similarities (lines 483-494). Please give more explanations for this.
The compounds identified by LC-MS are not presented in this paper (lines 1000 - 1002 mention that these appear in Table 3, but Table 3 is with "P-value from the ANOVA analysis".
The information from Conclusions must be improved.
Author Response
Response to Reviewer 1 Comments
Point 1. It is an interesting paper with a lot of experimental data, but a reorganization is necessary because it is hard to follow its.
Response 1. Thank you for your appreciation. We have moved some of the Tables (Table 2, 4) and Figures (Figures 7, 8, 17, part of Fig 16 and 18) to the Supplementary Materials and tried to explain better certain sections as requested. We decided to reorganize the paper as follows:
After optimization with RSM we introduced an additional subchapter 3.2 name which gathers all characterisations of the selected beverages and SeNPs presented, to be easier to follow and understand the redline of the paper.
Within this subchapter we reorganized the presentation in this order:
Silicon content (which has only one figure), composition of beverages with LC-MS, TPC, TFC, HAT, AOA of beverages (which are all related to each other), FTIR of beverages, SeNPs and standard compounds in order to characterise the “biocorona” of SeNPs (in this way we could justify the selection of standard molecules based on LC-MS, TPC etc presented before FTIR), SCOBY consortium modulation and finally the biological activities of SeNPs and beverages.
Figure 15 (now 13) was reorganized to occupy less space.
We hope that reducing the length of the main paper and the reorganization helped to make the paper easier to follow.
Point 2. It is not clear how help the soluble silicon content to obtain a high content of selenium. Please clarify this. Which is the correlation between title of paper and the discussions about silicon (see also paragraph from lines 577 - 587).
Response 2. The soluble silicon content does not help to obtain a higher amount of Se0, instead it was analysed because we previously reported in another study that the fermentation of pollen by a symbiotic culture of bacteria and yeasts (SCOBY/Kombucha) leads to the release of pollen content, including biosilica. The title is about Kombucha – pollen beverage enriched in SeNPs, therefore a thoroughly characterisation of the beverage was necessary, including the soluble Si released from the fermented pollen.
We also wanted to include this response in the experimental design to determine if soluble silicon content released from pollen is correlated in any way with the other responses. The Pearson correlation (Table 4) indicated that there was no significant correlation between the soluble silicon content and the other responses. Moreover, the p-value from the ANOVA analysis (Table 3) indicated that factor B – pollen significantly influences the silicon content of the beverage (p-value < 0.0001) and that there is also a significant interaction between the A-SCOBY and C-sodium selenite factors that influences the silicon content and that we tried to explain in the discussion section (lines 1275 – 1287). After the optimization, we selected the optimum and 2 other samples (as explained at (now) lines 502-510), resulting in the three samples (KPol5, KPol15 and KPol25) presented in Figure 11 (now Figure 4). These samples were prepared with the same amount of SCOBY and sodium selenite, the only difference being the concentration of pollen (5g in KPol5, 15g in KPol15, and 25g in KPol25), so Figure 4 indicates that a higher pollen concentration leads to the release of a higher amount of soluble silicon. We moved Figure 4 closer to the experimental design, as the first analysis of the selected variants, in order to be easier to follow the logic of the paper.
Point 3. In the sections 2.3.2 and 2.3.3, SeNPs not appears and is not clear how the experiments occurred related to these particles.
Response 3. We have added new explanations in the sections 2.3.2 (lines 304-306) and 2.3.3 (lines 312-318).
Point 4. It is not clear why the caffeic acid / hydroxycinnamic acids were chosen for FTIR similarities (lines 483-494). Please give more explanations for this.
Response 4. Caffeic acid IR spectrum was used to test the hypothesis of hydroxycinnamic acid as part of biocorona of the SeNPs, caffeic acid being one of the representative compounds of this subclass of aromatic acids. Polyphenols and proteins are known to be able to stabilize nanoparticles, including SeNPs. Following your enquiry, we performed additional FTIR analysis of other representative compounds, like catechin, epicatechin, and rutin as flavonoids representatives, and sinapic acid as additional hydroxycinnamic acid, gallic acid as polyphenol acid, and fatty acids like palmitic acid, all these spectra being presented as supplementary Figure S1. From all 8 representative compounds analysed, the highest similarities with SeNPs spectrum were observed for BSA (protein representative), gallic acid, and catechin, as presented in the new Figure 7b.
Point 5. The compounds identified by LC-MS are not presented in this paper (lines 1000 - 1002 mention that these appear in Table 3, but Table 3 is with "P-value from the ANOVA analysis".
Response 5. Thank you. It was an error. The compounds were presented in Table 6 (now Table 4). There were two captions of Table 3. We corrected the second Table 3 with Table 4 instead. We also corrected Table 3 with Table 4 in the Discussion Section, at lines 1000-1002 (now line 1163).
Point 6. The information from Conclusions must be improved.
Response 6. We revised the Conclusions section.
Reviewer 2 Report
A novel study is introduced to produce a selenium-fortified Kombucha-pollen beverage by in situ bio-synthesized selenium nanoparticles with high biocompatibility and antioxidant activity properties. The manuscript is well-written but extremely long. The experimental science reported seems sound, and the results support valid conclusions. Therefore, the manuscript may be considered for publication in ANTIOXIDANTS after the authors resolve minor observations such as the following:
Line 29, 102, 354, 367, 611, etc. Check for any missing spaces along the document.
Lines 125, 137, 139, 253, etc. Use “min” as a unit for minutes; “h” for hours in multiple occasions throughout the manuscript.
Line 260 Use letters instead numbers e.g., “two”
Line 486, 903, 1142, etc. Use consistently “biocorona” in italics as defined previously in line 479
Lines 1028-1032 A reference is missing
If the presence of resveratrol and other non-reported secondary metabolites were found evidently and definitely for the first time, they should also be mentioned in the abstract and conclusions. Otherwise, they may be reserved for another study or short note.
Line 1044 Write Malus in italics
Line 1142 Revise sentence i.e., polyphenols as a plural or single name
English is fine. Minimum corrections should be done
Author Response
Point 1. A novel study is introduced to produce a selenium-fortified Kombucha-pollen beverage by in situ bio-synthesized selenium nanoparticles with high biocompatibility and antioxidant activity properties. The manuscript is well-written but extremely long. The experimental science reported seems sound, and the results support valid conclusions. Therefore, the manuscript may be considered for publication in ANTIOXIDANTS after the authors resolve minor observations such as the following:
Line 29, 102, 354, 367, 611, etc. Check for any missing spaces along the document.
Response 1. We thank you very much for your description of our study. We also thank you for your observation. The missing spaces were revised throughout the manuscript. We reorganise the manuscript, reducing its length.
Point 2. Lines 125, 137, 139, 253, etc. Use “min” as a unit for minutes; “h” for hours in multiple occasions throughout the manuscript.
Response 2. Thank you. We modified.
Point 3. Line 260 Use letters instead numbers e.g., “two”
Response 3. Thank you. We used „two” instead of „2”.
Point 4. Line 486, 903, 1142, etc. Use consistently “biocorona” in italics as defined previously in line 479
Response 4. The term “biocorona” has been revised throughout the manuscript.
Point 5. Lines 1028-1032 A reference is missing
Response 5. Thank you for your observation. We decided to remove 6-methyl flavone, as one of compound previously not reported for which we could not provide 100% confident identification. Therefore, we removed this Information too.
Point 6. If the presence of resveratrol and other non-reported secondary metabolites were found evidently and definitely for the first time, they should also be mentioned in the abstract and conclusions. Otherwise, they may be reserved for another study or short note.
Response 6. Thank you for your valuable comment. We did a more in-depth survey of literature and found two other papers that reported resveratrol in black/green tea. We included these references in the paper. Nevertheless, after rechecking the LC-MS data, we realized that the m/z was between resveratrol and dihydro-resveratrol (ionized forms). Therefore, we additionally confirmed and quantified trans-resveratrol with HPLC-DAD. This does not completely exclude the presence of dihydro-resveratrol as well, therefore we let it as an open subject. All this Information was included in the paper. We rechecked the LC-MS data and reassigned some m/z values to other compounds (that have been previously reported in similar samples / situations), after some more in-depth investigation. We also changed the content of the Discussion. The compounds that were not previously reported (6-methyl flavone and p-coumaroylquinic acid) were also excluded. We shall continue the investigation of compositions in another study. The LC-MS data were moved at lines 525-530. The LC-MS data rea now before the FTIR of SeNPs in order to better explain the choice of representative compounds that could form the “biocorona”.
Point 7. Line 1044 Write Malus in italics
Response 7. Thank you for your observation. We modified.
Point 8. Line 1142 Revise sentence i.e., polyphenols as a plural or single name
Response 8. Thank you. We revised the sentence.